# EFFICIENT IMITATION UNDER MISSPECIFICATION

**Nicolas Espinosa-Dice**[*]**, Sanjiban Choudhury, Wen Sun**
Department of Computer Science
Cornell University
{ne229,sc2582,ws455}@cornell.edu

**Gokul Swamy**
Robotics Institute
Carnegie Mellon University
gswamy@cmu.edu

## ABSTRACT

We consider the problem of imitation learning under *misspecification*: settings where the learner is fundamentally unable to replicate expert behavior everywhere. This is often true in practice due to differences in observation space and action space expressiveness (e.g. perceptual or morphological differences between robots and humans). Given the learner must make some mistakes in the misspecified setting, interaction with the environment is fundamentally required to figure out which mistakes are particularly costly and lead to *compounding errors*. However, given the computational cost and safety concerns inherent in interaction, we'd like to perform as little of it as possible while ensuring we've learned a strong policy. Accordingly, prior work has proposed a flavor of *efficient inverse reinforcement learning* algorithms that merely perform a computationally efficient *local search* procedure with strong guarantees in the realizable setting. We first prove that under a novel structural condition we term *reward-agnostic policy completeness*, these sorts of local-search based IRL algorithms are able to avoid compounding errors. We then consider the question of *where* we should perform local search in the first place, given the learner may not be able to "walk on a tightrope" as well as the expert in the misspecified setting. We prove that in the misspecified setting, it is beneficial to *broaden* the set of states on which local search is performed to include those reachable by good policies the learner can actually play. We then experimentally explore a variety of sources of misspecification and how *offline* data can be used to effectively broaden where we perform local search from.

## 1 INTRODUCTION

Interactive imitation learning (IL) is a powerful paradigm for learning to make sequences of decisions from an expert demonstrating how to perform a task. While offline imitation learning approaches suffer from *covariate shift* between the training distribution (i.e. the expert's state distribution) and the test distribution (i.e. the learner's state distribution)—and the associated compounding of errors—interactive approaches allow the learner to observe states from the test distribution during training via performing rollouts, unlocking the ability to recover from mistakes (Ross et al., 2011).

Broadly speaking, the difference between the expert's performance and the learned policy's performance can be attributed to three forms of error:

1. *Optimization error*: The error resulting from imperfect search within a policy class (e.g. due to the difficulty of finding a global minima on non-convex problems).

2. *Finite sample error*: The statistical error arising from limited expert demonstrations.

3. *Misspecification error*: The irreducible error resulting from the learner's policy class not containing the expert's policy.

Notably, misspecification error is a function of both the expert policy and the learner's policy class; it cannot be improved by a better algorithm or more computation. Much of the prior work in imitation learning has focused on the first two sources of error, while avoiding the misspecification

---

[*]Correspondence to: Nicolas Espinosa-Dice <ne229@cornell.edu>.

error by imposing *expert realizability*: the assumption that the expert policy is within the learner's policy class (Kidambi et al., 2021; Swamy et al., 2021a; 2022b; Xu et al., 2023; Swamy et al., 2023; Ren et al., 2024). In other words, expert realizability is the assumption that the learner can perfectly imitate the expert's actions in *all* states: that more data and computation are all we need for optimality.

However, in practice, it is often impossible to imitate the expert perfectly. For example, realizability can also be an inaccurate assumption due to a mismatch between learner and expert action spaces. In humanoid robotics, the morphological differences between robots and humans prevent perfect human-to-robot motion re-targeting (Zhang et al., 2024; He et al., 2024; Al-Hafez et al., 2023). More generally, physical robots face the problem of changing dynamics due to wear and tear, as well as manufacturing imperfections, that cause variations in link lengths and other physical properties over the course of their operation. Thus, even if demonstrations are collected via teleoperation of a robot of the same make and model, expert realizability doesn't necessarily hold.

In response, our paper considers the more general *misspecified* setting, where the learner is not *necessarily* capable of perfectly imitating the expert's behavior. We analyze how the misspecification error interrelates with the optimization and finite sample errors. More specifically, we ask:

> *Under what condition can interactive imitation learning in the misspecified*
> *setting avoid compounding errors while retaining computational efficiency?*

The last two words of the preceding question are central to our study. By reducing the problem of imitation learning to reinforcement learning against a learned reward, interactive imitation learning approaches like inverse reinforcement learning (IRL; Ziebart et al. (2008); Ho & Ermon (2016)) face a similar *global* exploration problem to that of reinforcement learning—in the worst case, needing to explore all paths through the state space to find one reward (e.g. a tree-structured problem with sparse rewards) (Kakade, 2003; Swamy et al., 2023). Thus, in order to focus the exploration on useful states, efficient imitation algorithms leverage the expert's state distribution. In particular, rather than resetting the learner to the true starting state distribution, the learner is instead reset to states from the expert's demonstrations, resulting in an exponential decrease in interaction complexity (Swamy et al., 2023). We refer to this family of reset-based techniques as *efficient IRL*.

At a high level, efficient IRL can be understood as replacing the hard problem of global exploration with a local exploration problem *over the reset distribution*—in this case, the expert states. In other words, the RL subroutine can be thought of as optimizing a policy with a similar state visitation distribution to the expert. However, prior work in efficient IRL makes the crucial assumption expert realizability (Swamy et al., 2023). In the realizable setting, the best policy in the learner's policy class is the expert policy, so the goal of local policy search is to recover the expert policy, in which case it is natural to perform local policy optimization over the expert states.

However, in the misspecified setting, there may be no policies in the policy class that match the expert's state distribution, leading to two possible pitfalls of performing local policy search over expert states. First, while the learner may be able to optimize the policy *at* expert states (i.e. after being reset to them), the learner may not be able to *reach* the expert states. For example, even if a learned humanoid control policy can complete a backflip from the halfway point, it might not be able to get to the halfway point in the first place. Second, even if the learner can approximately match the expert's one-step local action, it may be unable to *follow-through* with the rest of the expert's trajectory. For example, even if a manipulator is able to move a peg close to a hole, it may be unable to insert the peg as dexterously as a person can due to a lack of haptic feedback.

Given the practical importance and relatively limited theoretical study of the misspecified setting, our first contribution is a condition that informs when IRL with expert resets works in the misspecified setting. At a high level, our condition measures how well, with respect to the globally optimal solution, the learner must perform local optimization over the expert states. Critically, expert realizability is not required for our condition to be satisfied, allowing for meaningful misspecification.

> **Contribution 1.** We define a new structural condition for the misspecified setting,
> *reward-agnostic approximate policy completeness*, under which our efficient IRL
> algorithm with expert resets can avoid quadratically compounding errors.

We then extend our analysis of efficient IRL to performing resets to states beyond those seen in the expert demonstration. As mentioned above, if no policy in the learner's policy class can reach an expert state or follow-through as the expert would, policy optimization at such a state doesn't seem particularly useful. This of course begs the question of where we should reset the learner to if we want to speed up the policy search subroutine of inverse RL if we can't uniformly imitate the expert.

Informally speaking, the choice of reset distribution in a local search procedure specifies the set of policies we're searching over: those with similar visitation distributions to the reset distribution. In the misspecified setting, our goal is to compete with the optimal *realizable* policy—the optimal policy the learner can actually choose. Thus, a natural choice for reset distribution is one that covers the states this optimal realizable policy visits, guaranteeing we're doing as well as one could hope.

Unfortunately, we don't know a priori what this optimal realizable policy is – we'd have already solved the misspecified imitation problem if we did. Intuitively though, one wants to *broaden* the support of the reset distribution to cover policies that aren't exactly the expert's. Practically speaking, we might get examples of such states by looking at *offline data*, such as robot play data (Lynch et al., 2020; Wang et al., 2023), suboptimal robot demonstrations (Brown et al., 2019; Chang et al., 2021; Yang et al., 2021; Hoang et al., 2024), or internet data (Chang et al., 2023; 2024), all of which are more likely to be realizable. We consider the question of how incorporating the offline data into the reset distribution affects IRL performance, and we show that the performance improvement depends on how well the new reset distribution covers the optimal realizable policy.

> **Contribution 2.** We theoretically show that broadening the reset distribution beyond the expert demonstrations so that it covers the state distribution of the optimal realizable policy, improves efficient IRL performance under misspecification.

Finally, we corroborate our theory by empirically investigating several potential sources of misspecification, showing that non-expert reset distributions are preferable under misspecification.

> **Contribution 3.** We explore several distinct forms of misspecification on continuous control and locomotion tasks, demonstrating that offline data that better covers the optimal realizable policy improves efficient IRL's performance.

We begin by defining the problem and postpone a discussion of related work to Appendix A.

## 2 IMITATION LEARNING IN THE MISSPECIFIED SETTING

**Notation.** We consider a finite-horizon Markov Decision Process (MDP), $\mathcal{M} = \langle \mathcal{S}, \mathcal{A}, P_h, r^\star, H, \mu \rangle$ (Puterman, 2014). $\mathcal{S}$ and $\mathcal{A}$ are the state space and action space, respectively. $P = \{P_h\}_{h=1}^H$ is the time-dependent transition function, where $P_h : \mathcal{S} \times \mathcal{A} \to \Delta(\mathcal{S})$ and $\Delta$ is the probability simplex. $r^\star : \mathcal{S} \times \mathcal{A} \to [0,1]$ is the ground-truth reward function, which is unknown, but we assume $r^\star \in \mathcal{R}$, where $\mathcal{R}$ is a class of reward functions such that $r : \mathcal{S} \times \mathcal{A} \to [0,1]$ for all $r \in \mathcal{R}$. $H$ is the horizon, and $\mu \in \Delta(\mathcal{S})$ is the starting state distribution. Let $\Pi = \{\pi : \mathcal{S} \to \Delta(\mathcal{A})\}$ be the class of stationary policies. We assume $\Pi$ and $\mathcal{R}$ are compact and convex. Let the class of non-stationary policies be defined by $\Pi^H = \{\pi_h : \mathcal{S} \to \Delta(\mathcal{A})\}_{h=1}^H$. A trajectory is given by $\xi = \{(s_h, a_h, r_h)\}_{h=1}^H$, where $s_h \in \mathcal{S}, a_h \in \mathcal{A}$, and $r_h = f(s_h, a_h)$ for some $f \in \mathcal{R}$. The distribution over trajectories formed by a policy is given by: $a_h \sim \pi(\cdot \mid s_h)$, $r_h = R_h(s_h, a_h)$, and $s_{h+1} \sim P_h(\cdot \mid s_h, a_h)$, for $h = 1, \ldots, H$. Let $d_{s_0,h}^\pi(s) = \mathbb{P}^\pi[s_h = s \mid s_0]$ and $d_{s_0}^\pi(s) = \frac{1}{H} \sum_{h=1}^H d_{s_0,h}^\pi(s)$.

**Misspecified Imitation.** As previous stated, much of the theoretical analysis in IL relies on the impractical assumption of a realizable expert policy (i.e. one that lies within the learner's policy class $\Pi$) (Kidambi et al., 2021; Swamy et al., 2021a;b; 2022a; Xu et al., 2023; Ren et al., 2024). In contrast to prior work, *we focus on the more realistic misspecified setting*, where the expert policy $\pi_E$ is not necessarily in the policy class $\Pi$. We consider a known sample of the expert policy's trajectories, where the dataset of state-action pairs sampled from the expert is $D_E = D_1 \cup D_2 \cup \ldots \cup D_H$, where $D_h = \{s_h, a_h\} \sim d_{\mu,h}^{\pi_E}$ and $|D_E| = N$. Let $\rho_h$ be a uniform distribution over the samples in $D_h$, and $\rho_E$ be a uniform distribution over the samples in $D_E$.

---

**Algorithm 1** Reset-Based IRL (Dual, Swamy et al. (2023))

---

1: **Input:** Expert state-action distributions $\rho_E$, policy class $\Pi$, reward class $\mathcal{R}$
2: **Output:** Trained policy $\pi$
3: **for** $i = 1$ to $N$ **do**
4:     `// No-regret step over rewards (e.g.  FTRL)`
5:     $r_i \leftarrow \arg\max_{r \in \mathcal{R}} J(\pi_E, r) - J(\text{Unif}(\pi_{1:i}), r)$
6:     `// Expert-competitive response by RL algorithm (e.g.  PSDP, Alg.  3)`
7:     $\pi_i \leftarrow \text{RL}(r = r_i, \rho = \rho_E)$
8: **end for**
9: **Return** $\pi_N$

---

**Goal of IRL.** We cast IRL as a Nash equilibrium computation (Syed & Schapire, 2007; Swamy et al., 2021a), where Sion's minimax theorem guarantees the existence of an equilibrium under the standard assumptions of compactness and convexity of the policy and reward classes. The ultimate objective of IRL is to learn a policy that matches expert performance. Because the ground-truth reward is unknown but belongs to the reward class (i.e. $r^\star \in \mathcal{R}$), we aim to learn a policy that performs well under *any* reward function in the reward class. For example, in problems like autonomous driving, $\mathcal{R}$ might include functions that capture staying close to the center of lanes and obeying speed limits. This is equivalent to finding the best policy under the *worst-case* reward (i.e. the reward function that maximizes the performance difference between the expert and learner). Formally, we find an equilibrium strategy for the game

$$\min_{\pi \in \Pi} \max_{r \in \mathcal{R}} J(\pi_E, r) - J(\pi, r), \tag{1}$$

where $J(\pi, r) = \mathbb{E}_{\xi \sim \pi}\left[\sum_{t=0}^{T} r(s_t, a_t)\right]$. By the compact and convex assumptions on $\Pi$ and $\mathcal{R}$, Sion's minimax theorem guarantees that the Nash equilibrium exists.

**IRL Taxonomy.** IRL algorithms consist of two steps: a reward update and a policy update. In the reward update, a *discriminator* is learned with the aim of differentiating the expert and learner trajectories—this is effectively trained as a classifier between the expert and learner trajectories. The policy is then optimized by an RL algorithm, with reward labels from the discriminator. IRL algorithms can be classified into *primal* and *dual* variants (Swamy et al., 2021a), the latter of which we use in our paper. An example dual algorithm is shown in Algorithm 1. In dual IRL algorithms, the discriminator is updated slowly via a no-regret step (e.g. Line 5, Follow The Regularized Leader McMahan (2011)), and the policy is updated via using an RL subroutine with reward labels $r$ (Ratliff et al., 2006; 2009; Ziebart et al., 2008; Swamy et al., 2021a).

**Reset Distribution.** RL algorithms require a reset distribution. Often, this is simply the MDP's starting state distribution, $\mu$ (Mnih, 2013; Schulman et al., 2015; 2017; Haarnoja et al., 2018). We differentiate between traditional IRL and efficient IRL by their RL subroutine's reset distribution, $\rho$. In traditional IRL algorithms, the reset distribution remains the true starting state distribution (i.e. $\rho = \mu$). In efficient IRL algorithms, the reset distribution is the expert's state distribution (i.e. $\rho = \rho_E$), which changes the RL subroutine from a best response step to an expert-competitive response (Swamy et al., 2023; Ren et al., 2024), while still maintaining performance guarantees. Intuitively, efficient IRL can be understood as replacing the *global* search problem inherent in RL with a *local* search over states from the demonstrations, thereby providing a computational speedup.

## 2.1 IMITATION UNDER MISSPECIFICATION IS HARD

Our paper considers efficient IRL in the misspecified setting, which begs the following question:

> *Is efficient IRL even possible in the misspecified setting without any assumptions on $\Pi$?*

We first consider *statistical efficiency*, which refers to the number of expert samples required to achieve strong performance guarantees, without any requirements on the *computational efficiency* required to do so. Observe that if statistically efficient imitation under misspecification is not possible, this implies that computationally efficient imitation is also not possible. Thankfully, we now prove that statistically efficient imitation is possible under misspecification with a novel algorithm.

**Statistically Efficient Imitation.** We present **S**cheffé **T**ournament **I**mitation **LE**arning (`STILE`), a statistically optimal algorithm in the misspecified setting. `STILE` assumes access to expert demonstrations (i.e. $D_E$) and known transition dynamics . `STILE`'s objective is to select the policy $\pi$ in the policy class whose induced state-action distribution (i.e. states and actions from $d^\pi$) is closest to the expert's empirical estimate (i.e. states and actions from $D_E$) for any bounded test function $f$. `STILE`'s procedure is defined in Appendix E, and we present the sample complexity for the misspecified setting below.[1]

**Theorem 2.1** (Sample Complexity of `STILE` under Misspecification). *Assume $\Pi$ is finite, but $\pi_E \notin \Pi$. With probability at least $1 - \delta$, `STILE` learns a policy $\hat{\pi}$ such that:*

$$V^{\pi_E} - V^{\hat{\pi}} \leq \frac{3}{1-\gamma} \min_{\pi \in \Pi} \|d^\pi - d^{\pi_E}\|_1 + \tilde{O}\left( \frac{1}{1-\gamma} \sqrt{\frac{\ln(|\Pi|) + \ln(1/\delta)}{N}} \right) \tag{2}$$

As shown in Theorem 2.1, `STILE` achieves a statistically efficient sample complexity that is linear in the horizon.[2] However, a tournament algorithm requires comparing every *pair* of policies, which isn't feasible with policy classes like deep networks, making `STILE` computationally inefficient.

**Computationally Efficient Imitation.** While `STILE` achieves a statistically optimal guarantee in the misspecified setting, we ultimately desire an algorithm that is also computationally feasible and can be implemented with standard policy classes used in deep reinforcement learning. To that end, we now consider computational efficiency, specifically focusing on *interaction sample efficiency*— the number of environment interactions needed to obtain strong performance. We consider an algorithm computationally efficient if its interaction complexity is polynomial in the MDP horizon. We present a lower bound showing that without any further assumptions, computationally efficient IRL under misspecification is not possible – i.e. a clear "no" to the question we began this section with.

**Theorem 2.2** (Lower Bound on Misspecified RL with Expert Feedback (Jia et al., 2024)). *For any $H \in \mathbb{N}$ and $C \in [2^H]$, there exists a policy class $\Pi$ with $|\Pi| = C$, expert policy $\pi_E \notin \Pi$, and a family of MDPs $\mathcal{M}$ with state space $\mathcal{S}$ of size $O(2^H)$, binary action space, and horizon $H$ such that any algorithm that returns a 1/4-optimal policy must either use $\Omega(C)$ queries to the expert oracle $O_{exp} : \mathcal{S} \times \mathcal{A} \to \mathcal{R}$, which returns $Q^{\pi_E}(s, a)$ (i.e. the Q value of expert policy $\pi_E$), or $\Omega(C)$ queries to a generative model, which allows the learner to query the transition and reward associated with a state-action pair on any state.[3]*

From Theorem 2.2, we establish that polynomial sample complexity in the misspecified IRL setting, where $\pi_E \notin \Pi$, cannot be guaranteed. In other words, ***computationally efficient IRL is not possible in the setting where no structure is assumed on the MDP***, even with access to a queryable expert policy like DAgger (Ross et al., 2011). The results show that while *statistically* efficient IRL in the misspecified setting is possible, *computationally* efficient IRL isn't without assuming additional structure, raising the question of what assumptions suffice to design practical algorithms.

## 3 APPROXIMATE POLICY COMPLETE INVERSE REINFORCEMENT LEARNING

We answer the question from the preceding section through an extension of *policy completeness*— a condition used in the analysis of policy gradient RL algorithms—and a corresponding efficient algorithm. We then show that, without an exponential amount of computation, our algorithm avoids quadratically compounding errors under approximate policy completeness.

### 3.1 APPROXIMATE POLICY COMPLETENESS

At a high-level, we aim to measure the *flexibility* of the policy class in the misspecified setting—a more nuanced metric than the simple binary of whether the expert policy lies in the class. While the realizability condition is an action level measure—it requires the learner be able to exactly match

---

[1]We present the sample complexity for the $\pi_E \in \Pi$ case in Appendix E.

[2]For convenience in the analysis, we consider an infinite horizon MDP.

[3]The generative model strictly generalizes the online interaction model, which is limited to playing actions in sequential states in a trajectory.

the expert's actions—policy completeness is a reward-sensitive measure. Intuitively, the policy completeness condition requires that the policy class contain a policy that can achieve comparable performance to the expert policy—*without* necessarily matching the exact actions of the optimal (i.e. expert) policy. Importantly, the policy completeness condition of RL algorithms depends on the MDP's reward function, which in the inverse reinforcement learning setting is unknown and is instead learned throughout training. In response, we introduce *reward-agnostic policy completeness*, the natural generalization of policy completeness extended to the imitation learning setting.

**Definition 3.1** (Reward-Indexed Policy Completeness Error). *Given the expert's state distribution* $\rho_E$, *MDP* $\mathcal{M}$ *with policy class* $\Pi$ *and reward class* $\mathcal{R}$, *learned policy* $\pi_i$, *and learned reward function* $r_i$, *define the reward-indexed policy completeness error of* $\mathcal{M}$ *to be*

$$\epsilon_\Pi^{\pi_i, r_i} := \mathbb{E}_{s \sim \rho_E} \left[ \max_{a \in \mathcal{A}} A_{r_i}^{\pi_i}(s, a) \right] - \max_{\pi' \in \Pi} \mathbb{E}_{s \sim \rho_E} \mathbb{E}_{a \sim \pi'(\cdot|s)} \left[ A_{r_i}^{\pi_i}(s, a) \right]. \tag{3}$$

We first present *reward-indexed policy completeness error*, which measures the policy class's ability to approximate the maximum possible advantage over the current policy. Intuitively, we can think of the second term as the learner's ability to improve the policy based on its policy class, since we consider the maximum over *policies in the learner's policy class*. The first term measures the maximum possible improvement over the current policy by taking the maximum over *all possible actions*, including the actions not realizable by the policies in the learner's policy class. We drop the $\rho_E$ notation when it is clear from context.

Because the learned policies and reward functions are not fixed throughout the algorithm (i.e. the policy and reward are updated each iteration), we consider the worst-case policy completeness error over all possible learned policies and reward functions in their respective classes. We define this worst-case policy completeness error as *reward-agnostic policy completeness error* below.

**Definition 3.2** (Reward-Agnostic Policy Completeness Error). *Given some expert state distribution* $\rho_E$ *and MDP* $\mathcal{M}$ *with policy class* $\Pi$ *and reward class* $\mathcal{R}$, *define the reward-agnostic policy completeness error of* $\mathcal{M}$ *to be*

$$\epsilon_\Pi^{\rho_E} := \max_{\pi \in \Pi, r \in \mathcal{R}} \epsilon_\Pi^{\pi, r, \rho_E} \tag{4}$$

Note that $0 \leq \epsilon_\Pi^{\pi_i, r_i} \leq \epsilon_\Pi \leq H$ for any $\pi_i \in \Pi$, $r_i \in \mathcal{R}$. Reward-agnostic policy completeness is therefore a measure of the policy class's ability to approximate the maximum possible advantage, over the expert's state distribution, under any reward function in the reward class. **We define the *approximate policy completeness* setting to be when $\epsilon_\Pi = \mathcal{O}(1)$.** Next, we will show that approximate policy completeness is sufficient for efficient IRL to avoid compounding errors.

## 3.2 EFFICIENT IRL WITH APPROXIMATE POLICY COMPLETENESS

We begin by presenting our efficient, reset-based IRL algorithm, **GU**iding **I**mi**T**aters with **A**rbitrary **R**esets (`GUITAR`). The high-level structure of our algorithm follows the standard efficient IRL procedure (Algorithm 1). The policy update reduces the global search problem of standard RL to local search by resetting the learner to some informative state distribution. The reward update is training a classifier between expert and learner trajectories. `GUITAR` is outlined in Algorithm 2.

**Policy Update.** More specifically, `GUITAR` employs Policy Search by Dynamic Programming (PSDP, Bagnell et al. (2003)), shown in Algorithm 3, for its strong theoretical guarantees. In practice, any RL algorithm, such as Soft Actor Critic (SAC, Haarnoja et al. (2018)), can be used. Crucially, in the policy update step, `GUITAR` replaces with standard reset distribution of its RL solver—both in theory and practice—with the expert's state distribution or some offline data distribution.[4]

**Reward Update.** `GUITAR` employs a no-regret update to the reward function. In theory, Online Mirror Descent (OMD, Nemirovskij & Yudin (1983)) is used due to its strong theoretical guarantees (Beck & Teboulle, 2003; Srebro et al., 2011). In practice, any no-regret update can be used, such as online gradient descent (Zinkevich, 2003) (i.e. taking a few gradient steps on the most recent data).

---

[4]Ren et al. (2024) established a reduction from inverse RL to expert-competitive RL. By replacing the reset distribution of the RL subroutine with expert states, the RL step in efficient IRL algorithms becomes an expert-competitive response, rather than a best-response step like in traditional IRL.

---

**Algorithm 2 GUiding ImiTaters with Arbitrary Resets (`GUITAR`)**

---

1: **Input:** Expert state-action distributions $\rho_E$, mixture of expert and offline state-action distributions $\rho_{\text{mix}}$, policy class $\Pi$, reward class $\mathcal{R}$
2: **Output:** Trained policy $\pi$
3: Set $\pi_0 \in \Pi$
4: **for** $i = 1$ to $N$ **do**
5:     Let
$$\hat{L}(\pi, r) = \mathbb{E}_{(s,a) \sim \rho_E} r(s, a) - \mathbb{E}_{(s,a) \sim d_\mu^\pi} r(s, a) \quad \text{\texttt{// Loss function}} \tag{5}$$
6:     Optimize
$$r_i = \texttt{OMD}(\pi_1, \ldots, \pi_{i-1}) \quad \text{\texttt{// Reward's no-regret update}} \tag{6}$$
7:     Optimize
$$\pi_i = \texttt{PSDP}(r = r_i, \rho = \rho_{\text{mix}}) \quad \text{\texttt{// Policy's expert-competitive response}} \tag{7}$$
8: **end for**
9: **Return** $\pi_i$ with lowest validation error

---

### 3.3 IS LOCAL SEARCH SUFFICIENT IN THE MISSPECIFIED SETTING?

`GUITAR` can be intuitively understood as reducing the hard problem of global RL search (i.e. exploration) to a local search problem over some state distribution. For simplicity, we assume this is the expert's state distribution in this section. At a high level, the local search problem can be understood as follows: working backwards from the final timestep, the learner is reset *to an expert state* and optimizes the policy (i.e. by selecting an action) to maximize the advantage over the current policy. Because the learner is reset to expert states, the performance guarantee of this local search is exclusively over the expert's state distribution. At test time, the learned policy's induced state distribution may deviate from the expert's policy because it may not be able to take the actions required to reach the expert state it saw during training. This is because of the policy class misspecification.

To answer the question of whether local search over expert states is sufficient in the misspecified setting, we leverage the approximate policy completeness condition. Approximate policy completeness condition measures how well the learner optimizes the policy in comparison to the globally optimal solution. We prove that ***under the approximate policy completeness condition, local search is sufficient to avoid quadratically compounding errors in the misspecified setting.***

**Theorem 3.3** (Sample Complexity of `GUITAR`). *Consider the case of infinite expert data samples, such that $\rho_E = d_\mu^{\pi_E}$. Denote $\pi_i = (\pi_{i,1}, \pi_{i,2}, \ldots, \pi_{i,H})$ as the policy returned by $\epsilon$-approximate PSDP at iteration $i \in [n]$ of `GUITAR`. Then,*

$$V^{\pi_E} - V^{\overline{\pi}} \leq \underbrace{H \epsilon_\Pi^{\rho_E}}_{\text{Misspecification Error}} + \underbrace{H^2 \epsilon}_{\text{Policy Optimization Error}} + \underbrace{H \sqrt{\frac{\ln |\mathcal{R}|}{n}}}_{\text{Reward Regret}}, \tag{8}$$

*where $H$ is the horizon, $n$ is the number of outer-loop iterations of the algorithm, and $\overline{\pi}$ is the trajectory-level average of the learned policies (i.e. $\pi_i$ at each iteration $i \in [n]$ of Algorithm 2).[5]*

**Misspecification Error.** The first term is the bound is due to misspecification. Unlike policy optimization error, the policy completeness error cannot be reduced with more environment interactions. Instead, it represents a fixed error that is a property of the MDP, the richness of the policy class, and the reward class. Prior work in efficient IRL ignored this error term by assuming expert realizability (Swamy et al., 2023). Notably, **`GUITAR`'s misspecification is linear in the horizon $H$**, thus avoiding the compounding errors that *all* offline algorithms like behavioral cloning suffer from.

**Policy Optimization Error.** The second term stems from the policy optimization error of the RL subroutine. This error can be interpreted as representing a tradeoff between environment interactions

---

[5] For clarity, we present `GUITAR`'s sample complexity in the infinite expert sample regime (i.e., when we have infinite samples from the expert policy, so $\rho_E = d_\mu^{\pi_E}$) – see Appendix H.2 for the finite sample analysis.

(i.e. computation) and error. It can be mitigated be decreasing the accuracy parameter $\epsilon$ of the RL solver—PSDP in this case. Set to $\epsilon = \frac{1}{H}$, the term is reduced to linear error in the horizon $H$. Crucially, our algorithm does not require global policy search. Because our algorithm uses PSDP over the expert's state distribution for the RL subroutine, **the policy optimization error can be reduced without requiring computation that scales exponentially in the task horizon $H$.**

**Reward Regret.** Finally, the last term, $H\sqrt{\frac{\ln|\mathcal{R}|}{n}}$, stems from the regret of the Online Mirror Descent update to the reward function. By the no-regret property and reward realizability, **we can reduce this term (to zero) by running more outer-loop iterations of GUITAR.**

To summarize, *GUITAR avoids compounding errors under APC without an exponential amount of computation, proving that local search is sufficient for efficient imitation under misspecification.*

## 4   THE THEORY OF WHERE TO SEARCH UNDER MISSPECIFICATION

In the preceding section, GUITAR replaces the RL subroutine's reset distribution with the expert states, which can be intuitively understood as reducing the global search problem of RL to a local search problem over the expert states. In other words, the reset distribution specifies where to perform local search. More formally, the RL solver, PSDP, learns a policy that competes against any policy covered by the reset distribution, so long as it is in the policy class (Bagnell et al., 2003). While in the preceding section we considered resetting to expert states, we might be unable to perfectly imitate the expert at all of these states due to misspecification. Roughly speaking, this can happen because the learner can't reach the expert states in the first place (e.g. they are through a narrow corridor the learner doesn't have the precision to pass through) or because the learner can't complete the rest of the episode the way the expert would (e.g. the learner can reach the same high velocity as the expert but loses the ability to avoid obstacles effectively). This begs the question:

*What reset distribution should we perform local search from in the misspecified setting?*

Intuitively, if we view the reset distribution as specifying the set of policies we compete against, we want to make sure our reset distribution covers the optimal *realizable* policy (i.e. the best choice the learner could actually make given restrictions on $\Pi$). More formally, we define

$$\pi^\star := \arg\min_{\pi \in \Pi} \max_{r \in \mathcal{R}} J(\pi_E, r) - J(\pi, r). \tag{9}$$

In words, $\pi^\star$ is the optimal policy *in* the learner's policy class against the worst-case reward function. By construction, the learner can actually reach $\pi^\star$'s states and follow through the way $\pi^\star$ would.

### 4.1   AUGMENTING THE RESET DISTRIBUTION WITH OFFLINE DATA

While resetting to $\pi^\star$'s state distribution seems promising for local policy search—the learner would then capable of replicating $\pi^\star$'s behavior—$\pi^\star$ is unknown a priori. Intuitively, what we'd like to do is *broaden* the set of states we've seen in the demonstrations so that we cover the states $\pi^\star$ visits during rollouts, rather than just covering the "tightrope" the expert walks on.

One potential source of this broader state distribution is *offline data*. In many practical applications, there is commonly an additional source of offline data, such as internet data (Chang et al., 2023; 2024), robot play data (Lynch et al., 2020; Wang et al., 2023), or suboptimal robot demonstrations (Brown et al., 2019; Chang et al., 2021; Yang et al., 2021; Hoang et al., 2024), often from a non-expert, realizable policy. Intuitively, we can think of the offline data as complementing the reset distribution of expert states with states that are reachable by the learner. In the next section, we will prove the condition under which resetting to this offline data benefits policy optimization.

More formally, we consider the general setting of having access to some offline dataset $D_{\text{off}} = \{s_i, a_i\}_{i=1}^M$, where $(s, a) \sim d_\mu^{\pi_B}$ and $\pi_B$ is some *realizable* behavior policy that is not necessarily as a high-quality as the expert $\pi_E$. Our approach for incorporating offline data into IRL is to augment the reset distribution of expert states with the offline data. This approach requires no change to the structure of GUITAR, in theory or in practice. We simply set the RL solver's reset distribution to the mixture of offline and expert states, $\rho = \rho_{\text{mix}}$, where we define $D_{\text{mix}} = D_E \cup D_{\text{off}}$ and $\rho_{\text{mix}}$ as the uniform distribution over $D_{\text{mix}}$. We weight the two distributions $\nu := \frac{N}{N+M} d_\mu^{\pi_E} + \frac{M}{N+M} d_\mu^{\pi_B}$,

while the reward update remains the same. The only modification to $\epsilon_\Pi$ is a change in the state distribution, replacing the distribution over expert samples, $\rho_E$, with the mixed distribution, $\rho_{\text{mix}}$.

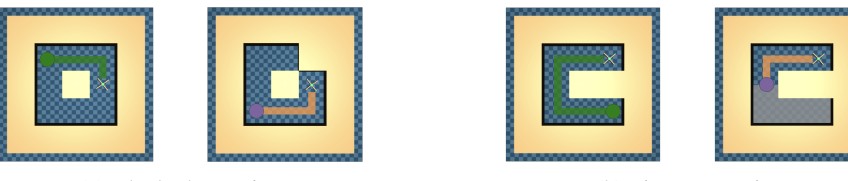

(a) Block obstruction           (b) Time constraint

Figure 1: **Maze construction under misspecification.** The expert's trajectory is shown in green, and the learner's trajectory is shown in orange. The green circle represents the goal position that returns the maximum *possible* reward, while the purple circle represents the goal position that returns the maximum *realizable* reward.

## 4.2 WHEN IS OFFLINE DATA BENEFICIAL IN IRL?

We can now precisely characterize when using offline data for resets is beneficial for IRL.

**Corollary 4.1** (Benefit of Offline Data). *If* $1 \le \left\| \frac{d_\mu^{\pi^\star}}{d_\mu^{\pi_B}} \right\|_\infty < \infty$, *incorporating offline data into the reset distribution improves the sample efficiency of* GUITAR *when*

$$C_B \left( \epsilon_\Pi^{\rho_{mix}} + \epsilon_\Pi^{\rho_{mix}} \sqrt{\frac{C_{\Pi,\mathcal{R}}}{N+M}} \right) < \epsilon_\Pi^{\rho_{mix}} + \epsilon_\Pi^{\rho_{mix}} \sqrt{\frac{C_{\Pi,\mathcal{R}}}{N}} \tag{10}$$

*where $N$ is the number of expert state-action pairs, $M$ is the number of offline state-action pairs, $C_{\Pi,\mathcal{R}} = \ln \frac{|\Pi||\mathcal{R}|}{\delta}$, and $C_B := \left\| \frac{d_\mu^{\pi^\star}}{d_\mu^{\pi_B}} \right\|_\infty$.*

Corollary 4.1[6] presents a sufficient condition for offline data improving GUITAR's sample efficiency *over* the algorithm with resets strictly to expert data. We observe that the benefit of offline data depends on how well the offline data covers the optimal realizable policy, $\pi^\star$, as well as the amount of expert and offline data. Intuitively, we can think of the coverage coefficient $C_B$ as the "exchange rate," measuring how useful the offline data is in comparison to the optimal realizable policy. This can be thought of intuitively as requiring that if $\pi^\star$ visits a state, $\pi_B$ does too, but not necessarily with an equal visitation frequency. We pay in terms of performance based on the mismatch in frequency. When the offline data covers $\pi^\star$'s state distribution well, $C_B$ is small, so the offline data helps.

Corollary 4.1 suggests that **the optimal offline data is one that covers the optimal *realizable* policy** $\pi^\star$, which further implies that the optimal reset distribution for efficient IRL under misspecification is that which best covers the covers the state distribution of $\pi^\star$. Practically, this means that when choosing what offline data and reset distribution to use, it may be more effective to use one from a *realizable* policy (e.g. a large amount of sub-optimal demonstrations from a robot of the same morphology) than data from an *unrealizable* expert policy. In the next section, we continue investigating the question of the optimal reset distribution empirically.

## 5 THE PRACTICE OF WHERE TO SEARCH UNDER MISSPECIFICATION

The results from Section 4 suggest that, in theory, the optimal reset distribution is one that best covers the state distribution of the optimal realizable policy, $\pi^\star$. In this section, we corroborate those theoretical findings with empirical results testing the effect of different reset distributions on IRL's performance in misspecified settings. In particular, we consider the case of misspecification due to unreachable expert states. Intuitively, if the learner cannot reach these states, policy optimization at such states is at best waste of computation and can potentially induce suboptimal behavior at states the learner actually *can* reach.[7]

---

[6]We present the complete finite sample analysis in Appendix H.

[7]Due to space limitations, we postpone our analysis of other misspecified settings to Appendix B.

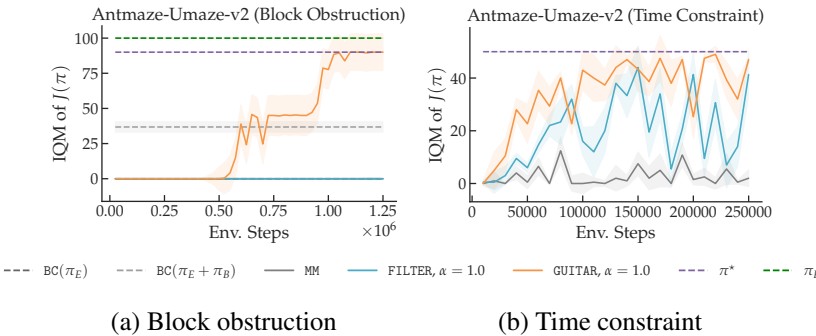

(a) Block obstruction       (b) Time constraint

Figure 2: **Unreachable expert states.** The reset distribution dictates where the learner performs local policy optimization. By resetting the learner to expert states, FILTER optimizes on the expert's state distribution. However, the learner is unable to reach certain expert states, so intuitively, there is no value for the learner in optimizing at those unreachable states. In contrast, all of the states in GUITAR's reset distribution are reachable by the learner and thus valuable for optimization. $\pi_E$ is the unrealizable expert policy, and $\pi^\star$ is the optimal realizable policy. Standard errors are computed across 10 seeds for Experiment (a) and 5 seeds for Experiment (b).

**Practical Motivation.** When learning to perform a particularly agile maneuver like dunking a basketball (He et al., 2025), a humanoid robot might benefit from being reset close to the rim in simulation to learn the skill of placing the ball into the hoop. However, the robot might lack the actuation capabilities to vertically jump in the same manner a person would, making these states unreachable. In such a setting, one would hope to learn an alternative strategy like a layup.

**Algorithm and Baselines.** For the experiments in this section, we compare our algorithm, GUITAR, against two behavioral cloning baselines (Pomerleau, 1988) and two IRL baselines (Swamy et al., 2023; 2021a). One behavioral cloning baseline is trained exclusively on the expert data, BC($\pi_E$), and the second is trained with both expert and offline data, BC($\pi_E + \pi_B$), when offline data is available. We consider a traditional IRL algorithm, MM, and an efficient IRL algorithm, FILTER (Swamy et al., 2021a; 2023). We train all IRL algorithms with the same expert data and hyperparameters. The difference between MM, FILTER, and GUITAR can be summarized by the reset distribution they use. MM resets the learner to the true starting state (i.e. $\rho = \mu$), while FILTER resets the learner to expert states (i.e. $\rho = \rho_E$). GUITAR resets the learner to offline data (i.e. $\rho = \rho_B$). For more details on the setup and implementations used in this section's experiments, refer to Appendix I.

**Experimental Setup.** To empirically investigate this setting, we consider a variant of the Antmaze-Umaze task, where a quadruped ant learns to solve a maze to reach a goal position (Fu et al., 2020). We impose two types of constraints to create misspecification. In the first variation (Figure 1a), we simulate unrealizable expert actions by placing a barrier in the learner's maze that prevents it from following the path of the expert. Notably, the block forces the learner to find an entirely different route through the maze. In the second variation (Figure 1b), we consider a "time constraint" that prevents the learner from reaching the second half of the maze. In this setting, we have access to expert data from an *unrealizable* expert policy, $\pi_E$, and offline data from the optimal *realizable* policy, $\pi^\star$.

**Experimental Results.** From Figure 2b, we see that focusing the reset distribution on the realizable policy $\pi^\star$ speeds up learning, as shown by GUITAR's improvement over FILTER. Interestingly, when the extent of the misspecification increases, the improvement exhibited by GUITAR over the baselines does too, as shown by Figure 2a. From Figure 2a, we observe that GUITAR is the only interactive algorithm capable of solving the hard exploration problem in the strongly misspecified setting, suggesting that the offline data—states from an optimal realizable policy—is a better reset distribution than expert states in this task. This agrees with what our preceding theory would predict.

Fundamentally, the reset distribution dictates where the learner performs local policy optimization, so by resetting the learner to expert states, FILTER optimizes on the expert's state distribution. However, the learner is unable to reach certain expert states, so intuitively, there is no value for the learner in optimizing at those unreachable states. In contrast, all of the states in GUITAR's reset distribution are reachable by the learner and thus valuable for optimization, improving performance.

ACKNOWLEDGMENTS

NED is supported by DARPA LANCER: LeArning Network CybERagents. SC is supported in part by Google Faculty Research Award, OpenAI SuperAlignment Grant, ONR Young Investigator Award, NSF RI #2312956, and NSF FRR#2327973. WS acknowledges funding from NSF IIS-#2154711, NSF CAREER #2339395, and DARPA LANCER: LeArning Network CybERagents. GKS was supported in part by an STTR grant. GKS thanks Drew Bagnell for several helpful discussions, particularly around how recoverability interacts with our results.

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

## A    Related Work

**Reinforcement Learning.** Prior work in reinforcement learning (RL) has examined leveraging exploration distributions to improve learning (Kakade & Langford, 2002; Bagnell et al., 2003; Ross et al., 2011; Song et al., 2022). Similar to Song et al. (2022), we consider access to offline data but differ by considering the imitation learning setting, while Song et al. (2022) considers the known-reward reinforcement learning setting. We adapt the Policy Search via Dynamic Programming (PSDP) algorithm of Bagnell et al. (2003) as our RL solver and leverage its performance guarantees in our analysis. We use Jia et al. (2024)'s lower bound on agnostic RL with expert feedback to show why agnostic IRL is hard.

Prior analyses of policy gradient RL algorithms—such as PSDP (Bagnell et al., 2003), Conservative Policy Iteration (CPI, Kakade & Langford (2002)), and Trust Region Policy Optimization (TRPO, Schulman et al. (2015))—use a *policy completeness* condition to establish a performance guarantee with respect to the *global*-optimal policy (Agarwal et al., 2019; Bhandari & Russo, 2024). In other words, policy completeness is used when comparing the learned policy to the optimal (i.e. best possible) policy and not simply the best policy in the policy class. We generalize the policy completeness condition from the RL setting with known rewards to the imitation learning setting with unknown rewards, resulting in novel structural condition we term reward-agnostic policy completeness. Our paper also builds on work in statistically tractable agnostic RL (Jia et al., 2024).

**Imitation Learning.** Our work examines the issue of distribution shift and compounding errors in IRL, which was introduced by Ross & Bagnell (2010). Ross et al. (2011)'s DAgger algorithm is capable of avoiding compounding errors but requires an interactive (i.e. queryable) expert and *recoverability* (Rajaraman et al., 2021; Swamy et al., 2021a), which we do not assume in our setting.

Our algorithm and results are not limited to the tabular and linear MDP settings, differentiating it from prior work in efficient imitation learning (Xu et al., 2023; Viano et al., 2024). Our work relates to Shani et al. (2022), who propose a Mirror Descent-based no-regret algorithm for online apprenticeship learning. We similarly use a mirror descent based update to our reward function, but differ from Shani et al. (2022)'s work by leveraging resets to expert and offline data to improve the interaction efficiency of our algorithm. Incorporating structured offline data has been proposed to learn reward functions in IRL (Brown et al., 2019; Brown & Niekum, 2019; Poiani et al., 2024), but rely on stronger assumptions about the structure of the offline data. In contrast, we do not use offline data in learning a reward function, instead using it to accelerate policy optimization via resets.

**Inverse Reinforcement Learning.** We build upon Swamy et al. (2023)'s technique of speeding up IRL by leveraging the expert's state distribution for learner resets. Our paper introduces the following key improvements to Swamy et al. (2023)'s work. First, while Swamy et al. (2023) relies on the impractical assumption of expert realizability, we tackle the more general, misspecified setting. Second, instead of assuming access to infinite expert data like Swamy et al. (2023), we consider the finite sample regime and further demonstrate how to incorporate offline data into IRL.

# B ADDITIONAL EMPIRICAL RESULTS UNDER MISSPECIFICATION

## B.1 MISSPECIFIED SETTING II: DIFFERENT DYNAMICS

Next, we consider the setting of misspecification due to a difference in dynamics between the expert demonstrations and the learner's environment (Sapora et al., 2024). This can make it difficult for the learner to "follow-through" like the expert would, complementing the setting we investigated above.

**Practical Motivation.** In humanoid robotics, the morphological differences between robots and humans prevent perfect human-to-robot motion retargeting (Zhang et al., 2024; He et al., 2024; Al-Hafez et al., 2023). As a result, the expert's and learner's dynamics may be different.

**Experimental Setup.** To empirically investigate this setting, we consider MuJoCo continuous control tasks where the expert and learner have different morphology, implemented by changing the link lengths and joint ranges between the expert—which uses the default values—and the learner. Changing the link lengths and joint ranges thereby changes the dynamics between the expert and learner. In such a setting, it is impossible to reset the learner to expert states without deforming the rigid body of the robot, and thus we cannot implement FILTER. For GUITAR, we reset the learner to states from the optimal realizable policy, $\pi^\star$, which has the same morphology as the learner.

**Experimental Results.** Figure 3 shows a speedup in learning enabled by using resets to states from a realizable policy, which GUITAR employs. In contrast, it is impossible to reset the learner to expert states, as FILTER does, due to the different morphology between the expert and learner. The consistently poor performance of the BC policies across the varying misspecified settings highlights the challenge of using offline data for direct policy learning, as the approach is sensitive to dynamics shifts in the misspecified setting. In contrast, the approach of using offline data for resets in interactive algorithms demonstrates stronger overall performance across tasks considered.

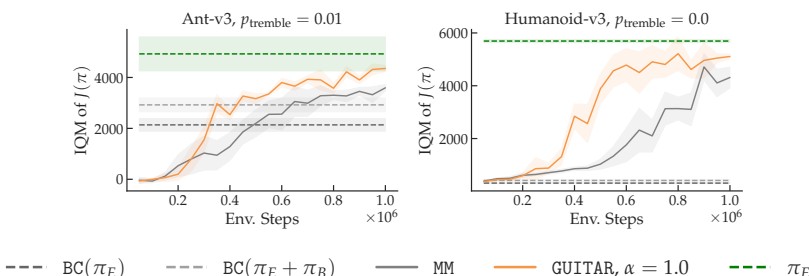

Figure 3: **Changing dynamics.** By resetting to states from a realizable policy, GUITAR shows a speedup over traditional inverse RL methods on a problem where the misspecification is due to a dynamics mismatch. Standard errors are computed across 5 seeds.

## B.2 ARE $\pi^\star$ STATES THE UNIQUE OPTIMAL RESET DISTRIBUTION?

We have shown theoretical and empirical results suggesting that the optimal reset distribution for efficient IRL are states from the best realizable policy, $\pi^\star$. In this section, we consider the question of whether $\pi^\star$ is the *unique* optimal reset distribution. While theory suggests that the reset distribution must *cover* the state distribution of $\pi^\star$, we explore resetting to *subsets* of $\pi^\star$'s state distribution.

**Practical Motivation.** Compared to the relatively unconstrained motion through free space before picking up an object, the contact phase of the manipulation interaction might be significantly more challenging for the learner to get correct. Even if $\pi^\star$ is able to successfully pick up and then manipulate some object, we might not need to expend as much compute performing local search at certain states where a wider range of actions is sufficient to make progress on the task. [8]

**Experimental Setup.** To empirically investigate this setting, we consider the D4RL Antmaze-Large tasks, where the expert data comes from the D4RL dataset. For GUITAR, we

---

[8]We leave formalizing this notion – which seems related to the *density* of "good enough" actions under the learner's advantage function – as an interesting question for future work.

isolate all short trajectories ($D_{\text{short}}$) and all long trajectories ($D_{\text{long}}$) in the expert dataset. We compare GUITAR($D_{\text{short}}$) and GUITAR($D_{\text{long}}$), which reset to short and long trajectories respectively. For simplicity, we do not add any additional misspecification and consider the policy that generated $D_{\text{full}}$ to be realizable and thus $\pi^\star$.

**Experimental Results.** In Figure 4, we observe that using the short-trajectory dataset as the reset distribution performs comparably to the full expert dataset, suggesting that full coverage of the expert's state distribution is not necessary. However, as demonstrated by the long-trajectory dataset failing to solve the problem, there are also subsets of the expert's state distribution that adversely affect learning. Crucially, we observe a difference in the coverage of the reset distributions when visualized in Figure 5, with the short-trajectory dataset having more focused coverage (i.e. coverage closer to goals). The results in Figure 4 suggest that the state distribution of $\pi^\star$ is not the only optimal reset distribution and resetting to a subset of $\pi^\star$'s state distribution can be equally efficient.

We leave a more nuanced theoretical exploration of what makes particular subsets of $\pi^\star$ better to reset to than others as a question for future work.

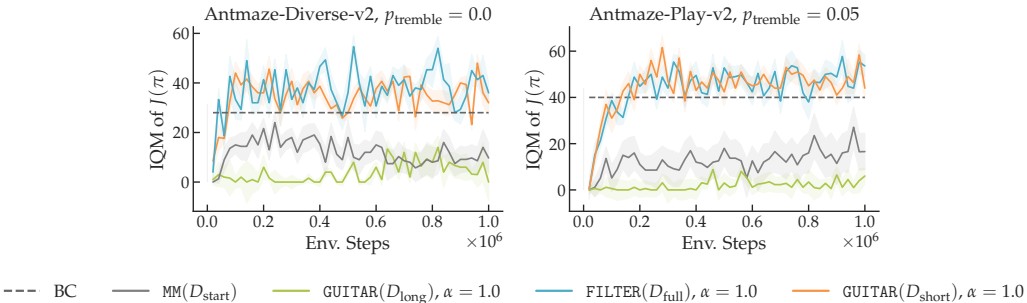

Figure 4: **Resets to subsets of $\pi^\star$'s state distribution.** We consider whether it is necessary, as the theory suggests, to reset to a distribution that covers $\pi^\star$'s state distribution. In this experiment, we reset to subsets of $\pi^\star$'s state distribution and compare their performance. The performance of GUITAR($D_{\text{short}}$) matches the performance of FILTER($D_{\text{full}}$), showing that full coverage of $\pi^\star$'s state distribution is not necessary in practice. Standard errors are computed across 10 seeds. During evaluation, agents sample a random action with probability $p_{\text{tremble}}$.

## C   DISCUSSION

In summary, we explore computationally efficient algorithms for imitation learning under misspecification, the setting where the learner cannot perfectly mimic the expert. In the misspecified setting, we demonstrate both theoretically and empirically that the lack of expert realizability affects the optimal choice of the reset distribution for the inner policy search step, thereby addressing one of the core limitations of Swamy et al. (2023). Notably, our analysis inherits a limitation from Swamy et al. (2023) and the work it builds on (Ross & Bagnell, 2014; Bagnell et al., 2003): it overlooks the fundamental ability of *any* policy to recover from mistakes. For example, on cliff-like problems, no policy can recover after a single mistake, but approximate policy completeness doesn't capture this.

The notion of *expert recoverability*—how effectively the expert can correct an arbitrary learner mistake—often appears in the analysis of queryable expert imitation learning algorithms like DAgger (Ross et al., 2011; Swamy et al., 2021a). Prior work has highlighted that, even for IRL algorithms, expert recoverability fundamentally dictates the ease of policy search (Swamy et al., 2021a, Section 4.3). While our notion of reward-agnostic policy completeness bears similarity to the variant of recoverability explored in Swamy et al. (2021a)—both impose bounds on advantages over possible reward functions—our definition is fundamentally off-policy in terms of the state distributions of the outer expectations. This off-policy nature makes it challenging to connect approximate policy completeness to the notion of a policy recovering from their *own* mistake. An interesting avenue for future work is to pursue a more refined notion of recoverability that inherits the strengths of both.

Furthermore, in the misspecified setting, where the learner cannot perfectly imitate the expert, it remains an open question whether the *expert*'s ability to recover from a mistake is the appropriate measure of problem difficulty. Thus, a more refined fusion of the above two concepts that also accounts for misspecification would be particularly illuminating for both theory and practice.

# D  MISSPECIFIED RL WITH EXPERT FEEDBACK

Theorem 2.2 establishes that polynomial sample complexity in the misspecified IRL setting, where $\pi_E \notin \Pi$, cannot be guaranteed. In other words, efficient IRL is not possible with no structure assumed on the MDP, even with access to a queryable expert policy like DAgger (Ross et al., 2011). Note that a generative model allows the learner to query the transition and reward associated with a state-action pair on any state, in contrast to an online interaction model that can only play actions on sequential states in a trajectory. For a more thorough discussion of their differences, see Kakade (2003).

More specifically, Theorem 2.2 presents a lower bound on agnostic RL with expert feedback. It assumes access to the true reward function and an expert oracle, $O_{\exp} : \mathcal{S} \times \mathcal{A} \to \mathcal{R}$, which returns $Q^{\pi_E}(s, a)$ for a given state-action pair $(s, a)$. The lower bound in Theorem 2.2 applies in the case where the expert oracle is replaced with a weaker expert action oracle (i.e. $\pi_E(s) : \mathcal{S} \to \mathcal{A}$) (Amortila et al., 2022; Jia et al., 2024). In agnostic IRL, we consider the even weaker setting of having a dataset of state-action pairs from the expert policy $\pi_E$.

Notably, this lower bound focuses on computational efficiency. Statistically efficient imitation *is* possible, and we present a statistically optimal imitation learning algorithm for the misspecified setting in Appendix E.

# E    STATISTICALLY OPTIMAL IMITATION UNDER MISSPECIFICATION

We begin with the following question: ignoring computation efficiency, what is the statistically optimal algorithm, with respect to the number of expert samples, for imitation learning in the misspecified setting?

We present **S**cheffé **T**ournament **I**mitation **LE**arning (STILE), a statistically optimal algorithm for the misspecified setting. For any two policies $\pi$ and $\pi'$, we denote by $f_{\pi,\pi'}$ the following witness function:

$$f_{\pi,\pi'} := \arg \max_{f:\|f\|_\infty \leq 1} \left[ \mathbb{E}_{s,a\sim d^\pi} f(s,a) - \mathbb{E}_{s,a\sim d^{\pi'}} f(s,a) \right], \tag{11}$$

and the set of witness functions as

$$\mathcal{F} = \{ f_{\pi,\pi'} : \pi, \pi' \in \Pi, \pi \neq \pi' \}. \tag{12}$$

Note that $|\mathcal{F}| \leq |\Pi|^2$. STILE selects $\hat\pi$ using the following procedure:

$$\hat\pi \in \arg \min_{\pi\in\Pi} \left[ \max_{f\in\mathcal{F}} \left( \mathbb{E}_{s,a\sim d^\pi} f(s,a) - \frac{1}{N} \sum_{i=1}^N f(s_i^*, a_i^*) \right) \right], \tag{13}$$

where $(s_i^*, a_i^*) \in D_E$. Notably, running a tournament algorithm requires comparing every pair of policies, which is not feasible with policy classes like deep neural networks, making STILE impractical to implement.

We present the analysis in the infinite-horizon setting for convenience.

**Theorem E.1** (Sample Complexity of STILE). *Assume $\Pi$ is finite and $\pi^\star \in \Pi$. With probability at least $1 - \delta$, STILE finds a policy $\hat\pi$*

$$V^{\pi_E} - V^{\hat\pi} \leq \frac{4}{1-\gamma} \sqrt{\frac{2\ln(|\Pi|) + \ln\left(\frac{1}{\delta}\right)}{N}}. \tag{14}$$

*Proof.* The proof relies on a uniform convergence argument over $\mathcal{F}$ of which the size is $|\Pi|^2$. First, note that for all policies $\pi \in \Pi$:

$$\max_{f\in\mathcal{F}} \left( \mathbb{E}_{s,a\sim d^\pi} f(s,a) - \mathbb{E}_{s,a\sim d^{\pi_E}} f(s,a) \right) = \max_{f:\|f\|_\infty\leq 1} \left( \mathbb{E}_{s,a\sim d^\pi} f(s,a) - \mathbb{E}_{s,a\sim d^{\pi_E}} f(s,a) \right) \tag{15}$$

$$= \|d^\pi - d^{\pi_E}\|_1 \tag{16}$$

where the first equality comes from the fact that $\mathcal{F}$ includes $\arg\max_{f:\|f\|_\infty\leq 1} \left[ \mathbb{E}_{s,a,s'\sim d^\pi} f(s,a) - \mathbb{E}_{s,a,s'\sim d^{\pi_E}} f(s,a) \right]$

Via Hoeffding's inequality and a union bound over $\mathcal{F}$, we get that with probability at least $1 - \delta$, for all $f \in \mathcal{F}$:

$$\left| \frac{1}{N} \sum_{i=1}^N f(s_i^*, a_i^*) - \mathbb{E}_{s,a\sim d^{\pi_E}} f(s,a) \right| \leq 2\sqrt{\frac{\ln(|\mathcal{F}|/\delta)}{N}} \tag{17}$$

$$:= \epsilon_{\text{stat}}. \tag{18}$$

Denote

$$\hat{f} := \arg\max_{f\in\mathcal{F}} \left[ \mathbb{E}_{s,a\sim d^{\hat\pi}} f(s,a) - \mathbb{E}_{s,a\sim d^{\pi_E}} f(s,a) \right] \tag{19}$$

and

$$\tilde{f} := \arg\max_{f\in\mathcal{F}} \mathbb{E}_{s,a\sim d^{\hat\pi}} f(s,a) - \frac{1}{N} \sum_{i=1}^N f(s_i, a_i). \tag{20}$$

Hence, for $\hat{\pi}$, we have:

$$\left\| d^{\hat{\pi}} - d^{\pi_E} \right\|_1 = \mathbb{E}_{s,a \sim d^{\hat{\pi}}} \hat{f}(s,a) - \mathbb{E}_{s,a \sim d^{\pi_E}} \hat{f}(s,a) \tag{21}$$

$$\leq \mathbb{E}_{s,a \sim d^{\hat{\pi}}} \hat{f}(s,a) - \frac{1}{N} \sum_{i=1}^{N} \hat{f}(s_i^*, a_i^*) + \epsilon_{\text{stat}} \tag{22}$$

$$\leq \mathbb{E}_{s,a \sim d^{\pi_E}} \tilde{f}(s,a) - \frac{1}{N} \sum_{i=1}^{N} \tilde{f}(s_i^*, a_i^*) + \epsilon_{\text{stat}} \tag{23}$$

$$\leq \mathbb{E}_{s,a \sim d^{\pi_E}} \tilde{f}(s,a) - \mathbb{E}_{s,a \sim d^{\pi_E}} \tilde{f}(s,a) + 2\epsilon_{\text{stat}} \tag{24}$$

$$= 2\epsilon_{\text{stat}} \tag{25}$$

where we use the optimality of $\hat{\pi}$ in the third inequality.

Recall that $V^\pi = \mathbb{E}_{s,a \sim d^\pi} r(s,a) / (1 - \gamma)$, so we have:

$$V^{\hat{\pi}} - V^{\pi_E} = \frac{1}{1 - \gamma} \left( \mathbb{E}_{s,a \sim d^{\hat{\pi}}} r(s,a) - \mathbb{E}_{s,a \sim d^{\pi_E}} r(s,a) \right) \tag{26}$$

$$\leq \frac{\sup_{s,a} |r(s,a)|}{1 - \gamma} \left\| d^{\hat{\pi}} - d^{\pi_E} \right\|_1 \tag{27}$$

$$\leq \frac{2}{1 - \gamma} \epsilon_{\text{stat}} \tag{28}$$

This concludes the proof. $\qquad\square$

### E.1 PROOF OF THEOREM 2.1

*Proof.* We first define some terms below. Denote $\tilde{\pi} := \arg\min_{\pi \in \Pi} \|d^\pi - d^{\pi_E}\|_1$. Let us denote:

$$\tilde{f} = \arg\max_{f \in \mathcal{F}} \left[ \mathbb{E}_{s,a \sim d^{\hat{\pi}}} f(s,a) - \mathbb{E}_{s,a \sim d^{\tilde{\pi}}} f(s,a) \right], \tag{29}$$

$$\bar{f} = \arg\max_{f \in \mathcal{F}} \left[ \mathbb{E}_{s,a \sim d^{\hat{\pi}}} f(s,a) - \frac{1}{N} \sum_{i=1}^{N} f(s_i^\star, a_i^\star) \right], \tag{30}$$

$$f' = \arg\max_{f \in \mathcal{F}} \left[ \mathbb{E}_{s,a \sim d^{\tilde{\pi}}} \left[ f(s,a) \right] - \frac{1}{N} \sum_{i=1}^{N} f(s_i^\star, a_i^\star) \right]. \tag{31}$$

Starting with triangle inequality, we have:

$$\left\| d^{\hat{\pi}} - d^{\pi^*} \right\|_1 \leq \left\| d^{\hat{\pi}} - d^{\tilde{\pi}} \right\|_1 + \left\| d^{\tilde{\pi}} - d^{\pi^*} \right\|_1 \tag{32}$$

$$= \mathbb{E}_{s,a \sim d^{\hat{\pi}}} \left[ \tilde{f}(s,a) \right] - \mathbb{E}_{s,a \sim d^{\tilde{\pi}}} \left[ \tilde{f}(s,a) \right] + \left\| d^{\tilde{\pi}} - d^{\pi^*} \right\|_1 \tag{33}$$

$$= \mathbb{E}_{s,a \sim d^{\hat{\pi}}} \left[ \tilde{f}(s,a) \right] - \frac{1}{N} \sum_{i=1}^{N} \tilde{f}(s_i, a_i^*) + \frac{1}{N} \sum_{i=1}^{N} \tilde{f}(s_i, a_i^*)$$

$$\quad - \mathbb{E}_{s,a \sim d^{\tilde{\pi}}} \left[ \tilde{f}(s,a) \right] + \left\| d^{\tilde{\pi}} - d^{\pi^*} \right\|_1 \tag{34}$$

$$\leq \mathbb{E}_{s,a \sim d^{\hat{\pi}}} \left[ \tilde{f}(s,a) \right] - \frac{1}{N} \sum_{i=1}^{N} \bar{f}(s_i, a_i^*) + \frac{1}{N} \sum_{i=1}^{N} \tilde{f}(s_i, a_i^*) - \mathbb{E}_{s,a \sim d^{\tilde{\pi}}} \left[ \tilde{f}(s,a) \right]$$

$$\quad + \left[ \mathbb{E}_{s,a \sim d^{\pi_E}} \tilde{f}(s,a) - \mathbb{E}_{s,a \sim d^{\tilde{\pi}}} \tilde{f}(s,a) \right] + \left\| d^{\tilde{\pi}} - d^{\pi^*} \right\|_1 \tag{35}$$

$$\leq \mathbb{E}_{s,a \sim d^{\tilde{\pi}}} \left[ f'(s,a) \right] - \frac{1}{N} \sum_{i=1}^{N} f'(s_i, a_i^*) + 2\sqrt{\frac{\ln(|\mathcal{F}|/\delta)}{N}} + 2 \left\| d^{\tilde{\pi}} - d^{\pi_E} \right\|_1 \tag{36}$$

$$\leq \mathbb{E}_{s,a \sim d^{\tilde{\pi}}} \left[ f'(s,a) \right] - \mathbb{E}_{s,a \sim d^{\pi_E}} \left[ f'(s,a) \right] + 4\sqrt{\frac{\ln(|\mathcal{F}|/\delta)}{N}} + 2 \left\| d^{\tilde{\pi}} - d^{\pi_E} \right\|_1 \tag{37}$$

$$\leq 3 \left\| d^{\pi_E} - d^{\tilde{\pi}} \right\|_1 + 4\sqrt{\frac{\ln(|\mathcal{F}|/\delta)}{N}}. \tag{38}$$

where the first inequality uses the definition of $\bar{f}$, the second inequality uses the fact that $\hat{\pi}$ is the minimizer of $\max_{f \in \mathcal{F}} \mathbb{E}_{s,a \sim d^\pi} f(s,a) - \frac{1}{N} \sum_{i=1}^{N} f(s_i^*, a_i^*)$. We also use Hoeffding's inequality where $\forall f \in \mathcal{F}$,

$$\left| \mathbb{E}_{s,a \sim d^{\pi_E}} f(s,a) - \sum_{i=1}^{N} f(s_i^*, a_i^*) \right| \leq 2\sqrt{\frac{\ln(|\mathcal{F}|/\delta)}{N}} \tag{39}$$

with probability at least $1 - \delta$. □

## F    FURTHER EXPLANATION OF GUITAR AND PSDP

---

**Algorithm 3** Policy Search Via Dynamic Programming (Bagnell et al., 2003)

---

1: **Input:** Reward function $r_i$, reset distribution $\rho$, and policy class $\Pi$
2: **Output:** Trained policy $\pi$
3: **for** $h = H, H - 1, \ldots, 1$ **do**
4:     Optimize

$$\pi_h \leftarrow \arg\max_{\pi' \in \Pi} \mathbb{E}_{s_h \sim \rho_h} \mathbb{E}_{a_h \sim \pi'(\cdot|s_h)} A_{r_i}^{\pi_{h+1}, \ldots, \pi_H}(s_h, a_h) \tag{40}$$

5: **end for**
6: **Return** $\pi = \{\pi_h\}_{h=1}^H$

---

**Algorithm 4** **GU**iding **I**mi**T**aters with **A**rbitrary **R**esets (GUITAR)

---

1: **Input:** Expert state-action distributions $\rho_E$, mixture of expert and offline state-action distributions $\rho_{\text{mix}}$, policy class $\Pi$, reward class $\mathcal{R}$
2: **Output:** Trained policy $\pi$
3: Set $\pi_0 \in \Pi$
4: **for** $i = 1$ to $N$ **do**
5:     Let

$$//\texttt{Loss function}$$
$$\hat{L}(\pi, r) = \mathbb{E}_{(s,a) \sim \rho_E} r(s,a) - \mathbb{E}_{(s,a) \sim d_\mu^\pi} r(s,a) \tag{41}$$

6:     Optimize

$$//\texttt{No-regret reward update}$$
$$r_i \leftarrow \arg\max_{r \in \mathcal{R}} \hat{L}(\pi_{i-1}, r) + \eta^{-1} \Delta_R(r \mid r_{i-1}) \tag{42}$$

7:     Optimize

$$//\texttt{Expert-competitive response with RL}$$
$$\pi_i \leftarrow \text{PSDP}(r = r_i, \rho = \rho_{\text{mix}}) \tag{43}$$

8: **end for**
9: **Return** $\pi_i$ with lowest validation error

---

The full IRL procedure is outlined in Algorithm 4. It can be summarized as (1) a no-regret reward update using Online Mirror Descent, and (2) an expert-competitive policy update using Policy Search by Dynamic Programming (PSDP) as the RL solver, where the learner is reset to a distribution $\rho$ in the RL subroutine.

Existing efficient IRL algorithms, such as MMDP (Swamy et al., 2023), reset the learner exclusively to expert states (i.e. the case where $\rho = \rho_E$). GUITAR can be seen as extending MMDP to a general reset distribution in the misspecified setting. We will focus on expert resets in the misspecified setting first, and we then consider other reset distributions in Section 4.

**Policy Update.** Following Ren et al. (2024)'s reduction of inverse RL to expert-competitive RL, we can use any RL algorithm to generate an expert-competitive response. We employ PSDP (Bagnell et al., 2003), shown in Algorithm 3, for its strong theoretical guarantees. In practice, any RL algorithm can be used, such as Soft Actor Critic (SAC, Haarnoja et al. (2018)).

**Reward Update.** We employ a no-regret update to the reward function. We employ Online Mirror Descent (Nemirovskij & Yudin, 1983; Beck & Teboulle, 2003; Srebro et al., 2011) for its strong theoretical guarantees, but in practice, any no-regret update can be used, such as gradient descent.

More specifically, the reward function is updated through Online Mirror Descent, such that

$$r_i \leftarrow \arg\max_{r \in \mathcal{R}} \hat{L}(\pi_{i-1}, r) + \eta^{-1} \Delta_R(r \mid r_{i-1}), \tag{44}$$

where $\Delta_R$ is the Bregman divergence with respect to the negative entropy function $R$. $\hat{L}(\pi, r)$ is the loss, defined by

$$\hat{L}(\pi, r) = \mathbb{E}_{(s,a) \sim \rho_E} r(s, a) - \mathbb{E}_{(s,a) \sim d_\mu^\pi} r(s, a), \tag{45}$$

with respect to the distribution of expert samples, $\rho_E$.

# G  PROOFS OF SECTION 3

## G.1  PROOF OF THEOREM 3.3

*Proof.* We consider the imitation gap of the expert and the average of the learned policies $\bar{\pi}$,

$$V^{\pi_E} - V^{\bar{\pi}} = \frac{1}{n} \sum_{i=1}^{n} \left( \mathbb{E}_{\zeta \sim \pi_E} \sum_{h=1}^{H} r^*(s,a) - \mathbb{E}_{\zeta \sim \pi_i} \sum_{h=1}^{H} r^*(s,a) \right) \tag{46}$$

$$= H \frac{1}{n} \sum_{i=1}^{n} \left( \mathbb{E}_{(s,a) \sim d_\mu^{\pi_E}} r^*(s,a) - \mathbb{E}_{(s,a) \sim d_\mu^{\pi_i}} r^*(s,a) \right) \tag{47}$$

$$= H \frac{1}{n} \sum_{i=1}^{n} L(\pi_i, r^*) \tag{48}$$

$$\leq H \frac{1}{n} \max_{r \in \mathcal{R}} \sum_{i=1}^{n} L(\pi_i, r) \tag{49}$$

$$\leq H \frac{1}{n} \max_{r \in \mathcal{R}} \sum_{i=1}^{n} \left( L(\pi_i, r) - L(\pi_i, r_i) + L(\pi_i, r_i) \right) \tag{50}$$

$$= H \frac{1}{n} \sum_{i=1}^{n} L(\pi_i, r_i) + H \frac{1}{n} \max_{r \in \mathcal{R}} \sum_{i=1}^{n} \left( L(\pi_i, r) - L(\pi_i, r_i) \right) \tag{51}$$

Applying the regret bound of Online Mirror Descent (Theorem K.2), we have

$$V^{\pi_E} - V^{\bar{\pi}} \leq H \frac{1}{n} \sum_{i=1}^{n} L(\pi_i, r_i) + H \sqrt{\frac{\ln |\mathcal{R}|}{n}} \tag{52}$$

$$= H \frac{1}{n} \sum_{i=1}^{n} \left( \frac{1}{H} \sum_{h=1}^{H} \mathbb{E}_{(s_h, a_h) \sim d_h^{\pi_E}} r_i(s_h, a_h) - \frac{1}{H} \sum_{h=1}^{H} \mathbb{E}_{(s_h, a_h) \sim d_h^{\pi_i}} r_i(s_h, a_h) \right)$$
$$+ H \sqrt{\frac{\ln |\mathcal{R}|}{n}} \tag{53}$$

$$= \frac{1}{n} \sum_{i=1}^{n} \left( \mathbb{E}_{s \sim \mu} V_{r_i}^{\pi_E} - \mathbb{E}_{s \sim \mu} V_{r_i}^{\pi_i} \right) + H \sqrt{\frac{\ln |\mathcal{R}|}{n}} \tag{54}$$

$$= \frac{1}{n} \sum_{i=1}^{n} \sum_{h=0}^{H-1} \left( \mathbb{E}_{(s_h, a_h) \sim d_h^{\pi_E}} A_{r_i, h}^{\pi_i}(s_h, a_h) \right) + H \sqrt{\frac{\ln |\mathcal{R}|}{n}} \tag{55}$$

Focusing on the interior summation, we have

$$\sum_{h=0}^{H-1} \mathbb{E}_{(s_h, a_h) \sim d_h^{\pi_E}} A_h^{\pi_i}(s_h, a_h) \leq \sum_{h=0}^{H-1} \mathbb{E}_{s_h \sim d_h^{\pi_E}} \max_{a \in \mathcal{A}} A_h^{\pi_i}(s_h, a) \tag{56}$$

$$= \sum_{h=0}^{H-1} \mathbb{E}_{s_h \sim d_h^{\pi_E}} \max_{a \in \mathcal{A}} A_h^{\pi_i}(s_h, a) - \epsilon_{\Pi, h}^{\rho_E} + \epsilon_{\Pi, h}^{\rho_E} \tag{57}$$

$$= \sum_{h=0}^{H-1} \max_{\pi' \in \Pi} \mathbb{E}_{s_h \sim d_h^{\pi_E}} \mathbb{E}_{a \sim \pi'(\cdot|s)} A_h^{\pi_i}(s_h, a) + \epsilon_{\Pi, h}^{\rho_E} \tag{58}$$

$$\leq H^2 \epsilon + H \epsilon_\Pi^{\rho_E} \tag{59}$$

where the last line holds by PSDP's performance guarantee (Bagnell et al., 2003).

Applying Equation 59 to Equation 55, we have

$$V^{\pi_E} - V^{\bar{\pi}} \le \frac{1}{n} \sum_{i=1}^{n} \sum_{h=0}^{H-1} \left( \mathbb{E}_{(s_h, a_h) \sim d_h^{\pi_E}} A_{r_i, h}^{\pi_i}(s_h, a_h) \right) + H\sqrt{\frac{\ln |\mathcal{R}|}{n}} \tag{60}$$

$$\le \frac{1}{n} \sum_{i=1}^{n} \left( H^2 \epsilon + H \epsilon_{\Pi}^{\rho_E} \right) + H\sqrt{\frac{\ln |\mathcal{R}|}{n}} \tag{61}$$

$$\le H^2 \epsilon + H \epsilon_{\Pi}^{\rho_E} + H\sqrt{\frac{\ln |\mathcal{R}|}{n}} \tag{62}$$

which completes the proof. $\square$

## H    PROOFS OF SECTION 4

### H.1    LEMMAS OF THEOREM H.5

**Lemma H.1** (Reward Regret Bound). *Recall that*

$$\hat{L}(\pi, r) = \mathbb{E}_{(s,a)\sim\rho_E} r(s,a) - \mathbb{E}_{(s,a)\sim d_\mu^\pi} r(s,a). \tag{63}$$

*Suppose that we update the reward via the Online Mirror Descent algorithm. Since $0 \le r(s,a) \le 1$ for all $s, a$, then $\sup_{\pi\in\Pi, r\in\mathcal{R}} \hat{L}(\pi, r) \le 1$. Applying Theorem K.2 with $B = 1$, the regret is given by*

$$\lambda_n = \sup_{r\in\mathcal{R}} \frac{1}{n} \sum_{i=1}^n \hat{L}(\pi_i, r) - \frac{1}{n} \sum_{i=1}^n \hat{L}(\pi_i, r_i) \tag{64}$$

$$\le \sqrt{\frac{2\ln|\mathcal{R}|}{n}} \tag{65}$$

$$= \sqrt{\frac{C_1}{n}}, \tag{66}$$

*where $C_1 = 2\ln|\mathcal{R}|$ and $n$ is the number of updates.*

**Lemma H.2** (Statistical Difference of Losses). *With probability at least $1 - \delta$,*

$$L(\pi, r) \le \hat{L}(\pi, r) + \sqrt{\frac{C}{N}}, \tag{67}$$

*where $C = \ln\frac{2|\mathcal{R}|}{\delta}$ and $N$ is the number of state-action pairs from the expert.*

*Proof.* By definition of $L$ and $\hat{L}$, for any $\pi \in \Pi$ and $r \in \mathcal{R}$, we have

$$\left| L(\pi, r) - \hat{L}(\pi, r) \right| = \left| \mathbb{E}_{(s,a)\sim d_\mu^{\pi_E}} r(s,a) - \mathbb{E}_{(s,a)\sim d_\mu^\pi} r(s,a) \right.$$
$$\left. - \left( \mathbb{E}_{(s,a)\sim\rho_E} r(s,a) - \mathbb{E}_{(s,a)\sim d_\mu^\pi} r(s,a) \right) \right| \tag{68}$$

$$= \left| \mathbb{E}_{(s,a)\sim d_\mu^{\pi_E}} r(s,a) - \mathbb{E}_{(s,a)\sim\rho_E} r(s,a) \right| \tag{69}$$

$$= \left| \mathbb{E}_{(s,a)\sim d_\mu^{\pi_E}} r(s,a) - \frac{1}{N} \sum_{(s_i,a_i)\in D_E}^N r(s_i, a_i) \right| \tag{70}$$

$$\le \sqrt{\frac{1}{2N} \ln\frac{2|\mathcal{R}|}{\delta}} \tag{71}$$

$$\le \sqrt{\frac{C}{N}}, \tag{72}$$

where $C = 4\ln\frac{2|\mathcal{R}|}{\delta}$. The fourth line holds by Hoeffding's inequality and a union bound. Specifically, we apply Corollary K.1 with $c = 1$, since all rewards are bounded by 0 and 1. We take a union bound over all reward functions in the reward class $\mathcal{R}$. Note that the terms involving $\pi$ cancel out, so the union bound only applies to the reward function class $\mathcal{R}$. Rearranging terms gives the desired bound. $\square$

**Lemma H.3** (Advantage Bound). *Suppose that $\epsilon = 0$ and reward function $r_i$ are the input parameters to PSDP, and $\pi_i = (\pi_1^i, \pi_2^i, \ldots, \pi_H^i)$ is the output learned policy. Then, with probability at least $1 - \delta$,*

$$\mathbb{E}_{s\sim d^{\pi_E}} \max_{a\in\mathcal{A}} A^{\pi_i}(s,a) \le \min\left\{ \epsilon_\Pi^{\rho_{mix}} + \epsilon_\Pi^{\rho_{mix}}\sqrt{\frac{C_0}{N}}, C_B\left( \epsilon_\Pi^{\rho_{mix}} + \epsilon_\Pi^{\rho_{mix}}\sqrt{\frac{C_0}{N+M}} \right) \right\} \tag{73}$$

*where $C_B = \left\| \frac{d_\mu^{\pi_E}}{d_\mu^{\pi_B}} \right\|_\infty$, $H$ is the horizon, $N$ is the number of expert state-action pairs, $M$ is the number of offline state-action pairs, and $C_0 = 2\ln\frac{|\Pi||\mathcal{R}|}{\delta}$.*

*Proof.* Suppose that $\epsilon = 0$ is the input accuracy parameter to PSDP, and the advantages are computed under reward function $r_i$. PSDP is guaranteed to terminate and output a policy $\pi_i = (\pi_1^i, \pi_2^i, \ldots, \pi_H^i)$, such that

$$H\epsilon \geq \max_{\pi' \in \Pi} \mathbb{E}_{s_h \sim \rho_{\text{mix},h}} \mathbb{E}_{a \sim \pi'(\cdot|s)} A_h^{\pi_i}(s_h, a) \tag{74}$$

for all $h \in [H]$ (Bagnell et al., 2003). Consequently, we have

$$H\epsilon \geq \max_{\pi' \in \Pi} \mathbb{E}_{s \sim \rho_{\text{mix}}} \mathbb{E}_{a \sim \pi'(\cdot|s)} A^{\pi_i}(s, a) \tag{75}$$

$$= \max_{\pi' \in \Pi} \mathbb{E}_{s \sim \rho_{\text{mix}}} \mathbb{E}_{a \sim \pi'(\cdot|s)} A^{\pi_i}(s, a) + \epsilon_{\Pi,r_i}^{\rho_{\text{mix}}} - \epsilon_{\Pi,r_i}^{\rho_{\text{mix}}} \tag{76}$$

$$= \mathbb{E}_{s \sim \rho_{\text{mix}}} \max_{a \in \mathcal{A}} A^{\pi_i}(s, a) - \epsilon_{\Pi,r_i}^{\rho_{\text{mix}}} \tag{77}$$

By definition, $0 \leq \epsilon_{\Pi,r_i}^{\rho_{\text{mix}}} \leq \epsilon_{\Pi}$, so for any $x \in \mathbb{R}$, $x - \epsilon_{\Pi,r_i}^{\rho_{\text{mix}}} \geq x - \epsilon_{\Pi}^{\rho_{\text{mix}}}$, so

$$H\epsilon \geq \mathbb{E}_{s \sim \rho_{\text{mix}}} \max_{a \in \mathcal{A}} A^{\pi_i}(s, a) - \epsilon_{\Pi}^{\rho_{\text{mix}}}. \tag{78}$$

Rearranging the terms gives us

$$\mathbb{E}_{s \sim \rho_{\text{mix}}} \max_{a \in \mathcal{A}} A^{\pi_i}(s, a) \leq H\epsilon + \epsilon_{\Pi}^{\rho_{\text{mix}}} \tag{79}$$

$$= \epsilon_{\Pi}^{\rho_{\text{mix}}}, \tag{80}$$

where the last line holds by our assumption that $\epsilon = 0$.

**Case 1: Jettison Offline Data.** We will first consider the case where offline data is useless, in which case we will focus on the expert data.

Note that $\max_{a \in \mathcal{A}} A^{\pi_i}(s, a) \geq 0$ for all $s \in \mathcal{S}$ and $h \in [H]$. Applying the definition of $\rho_{\text{mix}}$,

$$\mathbb{E}_{s \sim \rho_{\text{mix}}} \max_{a \in \mathcal{A}} A^{\pi_i}(s, a) = \mathbb{E}_{s \sim \rho_E} \max_{a \in \mathcal{A}} A^{\pi_i}(s, a) + \mathbb{E}_{s \sim \rho_B} \max_{a \in \mathcal{A}} A^{\pi_i}(s, a). \tag{81}$$

Consequently, we know that

$$\epsilon_{\Pi}^{\rho_{\text{mix}}} \geq \mathbb{E}_{s \sim \rho_E} \max_{a \in \mathcal{A}} A^{\pi_i}(s, a) \tag{82}$$

$$= \frac{1}{N} \sum_{s_i \in D_E}^{N} \max_{a \in \mathcal{A}} A^{\pi_i}(s_i, a) \tag{83}$$

Because $\max_{a \in \mathcal{A}} A^{\pi_i}(s, a) \geq 0$ for all $s \in \mathcal{S}$ and $a \in \mathcal{A}$, we know $\max_{a \in \mathcal{A}} A^{\pi_i}(s_i, a) \leq \epsilon_{\Pi}^{\rho_{\text{mix}}}$ for all $s_i \in D_E$. We apply Hoeffding's inequality (Corollary K.1) with $c = (\epsilon_{\Pi}^{\rho_{\text{mix}}})^2$ to bound the difference between $\mathbb{E}_{s \sim d_\mu^{\pi_E}} \max_{a \in \mathcal{A}} A^{\pi_i}(s, a)$ and $\mathbb{E}_{s \sim \rho_E} \max_{a \in \mathcal{A}} A^{\pi_i}(s, a)$. We apply a union bound on the policy and reward function. As stated previously, $\max_{a \in \mathcal{A}} A^{\pi_i}(s, a) \geq 0$ for all $s \in \mathcal{S}$. By Hoeffding's inequality, with probability $1 - \delta$, we have

$$\left| \mathbb{E}_{s \sim d_\mu^{\pi_E}} \max_{a \in \mathcal{A}} A^{\pi_i}(s, a) - \mathbb{E}_{s \sim \rho_E} \max_{a \in \mathcal{A}} A^{\pi_i}(s, a) \right| = \left| \mathbb{E}_{s \sim d_\mu^{\pi_E}} \max_{a \in \mathcal{A}} A^{\pi_i}(s, a) \right. \tag{84}$$

$$\left. - \frac{1}{N} \sum_{s_i \in D_E}^{N} \max_{a \in \mathcal{A}} A^{\pi_i}(s_i, a) \right| \tag{85}$$

$$\leq \sqrt{(\epsilon_{\Pi}^{\rho_{\text{mix}}})^2 \frac{1}{2N} \ln \frac{|\Pi||\mathcal{R}|}{\delta}} \tag{86}$$

$$\leq \epsilon_{\Pi}^{\rho_{\text{mix}}} \sqrt{\frac{C_0}{N}}, \tag{87}$$

where $C_0 = 2 \ln \frac{|\Pi||\mathcal{R}|}{\delta}$. Note that the cardinality of the set of advantage functions over all possible policies is upper bounded by the cardinalities of the policy and reward classes. Rearranging the terms and applying Equation 82 yields

$$\mathbb{E}_{s \sim d_\mu^{\pi_E}} \max_{a \in \mathcal{A}} A^{\pi_i}(s, a) \leq \epsilon_{\Pi}^{\rho_{\text{mix}}} + \epsilon_{\Pi}^{\rho_{\text{mix}}} \sqrt{\frac{C_0}{N}} \tag{88}$$

**Case 2: Leverage Offline Data.** Next, we consider the case where offline data is useful, specifically where there is good coverage of the expert data.

Next, we apply Hoeffding's inequality (Corollary K.1) to bound the difference between $\mathbb{E}_{s \sim \nu} \max_{a \in \mathcal{A}} A^{\pi_i}(s, a)$ and $\mathbb{E}_{s \sim \rho_{\text{mix}}} \max_{a \in \mathcal{A}} A^{\pi_i}(s, a)$. We apply a union bound on the policy and reward function. We use $c = \epsilon_\Pi^2$ for a similar argument to the one used in Case 1. As stated previously, $\max_{a \in \mathcal{A}} A^{\pi_i}(s, a) \geq 0$ for all $s \in \mathcal{S}$. By Hoeffding's inequality, with probability $1 - \delta$, we have

$$\left| \mathbb{E}_{s \sim \nu} \max_{a \in \mathcal{A}} A^{\pi_i}(s, a) - \mathbb{E}_{s \sim \rho_{\text{mix}}} \max_{a \in \mathcal{A}} A^{\pi_i}(s, a) \right| = \left| \mathbb{E}_{s \sim \nu} \max_{a \in \mathcal{A}} A^{\pi_i}(s, a) \right. \tag{89}$$

$$\left. - \frac{1}{N + M} \sum_{s_i \in D_{\text{mix}}^{N+M}} \max_{a \in \mathcal{A}} A^{\pi_i}(s_i, a) \right|$$

$$\leq \sqrt{(\epsilon_\Pi^{\rho_{\text{mix}}})^2 \frac{1}{2(N + M)} \ln \frac{|\Pi||\mathcal{R}|}{\delta}} \tag{90}$$

$$\leq \epsilon_\Pi^{\rho_{\text{mix}}} \sqrt{\frac{C_0}{N + M}} \tag{91}$$

where $C_0 = 2 \ln \frac{|\Pi||\mathcal{R}|}{\delta}$. Note that the cardinality of the set of advantage functions over all possible policies is upper bounded by the cardinalities of the policy and reward classes. Rearranging the terms and applying Equation 79 yields

$$\mathbb{E}_{s \sim \nu} \max_{a \in \mathcal{A}} A^{\pi_i}(s, a) \leq \epsilon_\Pi^{\rho_{\text{mix}}} + \epsilon_\Pi^{\rho_{\text{mix}}} \sqrt{\frac{C_0}{N + M}}. \tag{92}$$

By linearity of expectation, and using the fact that $1 \leq C_B < \infty$, we have

$$\mathbb{E}_{s \sim d^{\pi_E}} \max_{a \in \mathcal{A}} A^{\pi_i}(s, a) = \frac{N}{N + M} \mathbb{E}_{s \sim d^{\pi_E}} \max_{a \in \mathcal{A}} A^{\pi_i}(s, a)$$

$$+ \frac{M}{N + M} \mathbb{E}_{s \sim d^{\pi_E}} \max_{a \in \mathcal{A}} A^{\pi_i}(s, a) \tag{93}$$

$$\leq \frac{N}{N + M} \mathbb{E}_{s \sim d^{\pi_E}} \max_{a \in \mathcal{A}} A^{\pi_i}(s, a)$$

$$+ C_B \frac{M}{N + M} \mathbb{E}_{s \sim d^{\pi_B}} \max_{a \in \mathcal{A}} A^{\pi_i}(s, a) \tag{94}$$

$$\leq C_B \frac{N}{N + M} \mathbb{E}_{s \sim d^{\pi_E}} \max_{a \in \mathcal{A}} A^{\pi_i}(s, a)$$

$$+ C_B \frac{M}{N + M} \mathbb{E}_{s \sim d^{\pi_B}} \max_{a \in \mathcal{A}} A^{\pi_i}(s, a) \tag{95}$$

$$\leq C_B \mathbb{E}_{s \sim \nu} \max_{a \in \mathcal{A}} A^{\pi_i}(s, a) \tag{96}$$

Applying Equation 96 to Equation 92, we have

$$\mathbb{E}_{s \sim d^{\pi_E}} \max_{a \in \mathcal{A}} A^{\pi_i}(s, a) \leq C_B \mathbb{E}_{s \sim \nu} \max_{a \in \mathcal{A}} A^{\pi_i}(s, a) \tag{97}$$

$$\leq C_B \left( \epsilon_\Pi^{\rho_{\text{mix}}} + \epsilon_\Pi^{\rho_{\text{mix}}} \sqrt{\frac{C_0}{N + M}} \right) \tag{98}$$

**Final Result.** Using the bounds from Case 1 and Case 2, we have

$$\mathbb{E}_{s \sim d^{\pi_E}} \max_{a \in \mathcal{A}} A^{\pi_i}(s, a) \leq \min \left\{ \epsilon_\Pi^{\rho_{\text{mix}}} + \epsilon_\Pi^{\rho_{\text{mix}}} \sqrt{\frac{C_0}{N}}, \right. \tag{99}$$

$$\left. C_B \left( \epsilon_\Pi^{\rho_{\text{mix}}} + \epsilon_\Pi^{\rho_{\text{mix}}} \sqrt{\frac{C_0}{N + M}} \right) \right\} \tag{100}$$

where $C_B = \left\| \frac{d_\mu^{\pi_E}}{d_\mu^{\pi_B}} \right\|_\infty$, $H$ is the horizon, $N$ is the number of expert state-action pairs, $M$ is the number of offline state-action pairs, and $C_0 = 2\ln\frac{|\Pi||\mathcal{R}|}{\delta}$. $\qquad\qquad\square$

**Lemma H.4** (Loss Bound). *Suppose that $\epsilon = 0$ and reward function $r_i$ are the input parameters to PSDP, and $\pi_i = (\pi_1^i, \pi_2^i, \ldots, \pi_H^i)$ is the output learned policy. Then, with probability at least $1 - \delta$,*

$$\hat{L}(\pi_i, r_i) \leq \min\left\{ \epsilon_\Pi^{\rho_{mix}} + \epsilon_\Pi^{\rho_{mix}}\sqrt{\frac{C_0}{N}}, C_B\left(\epsilon_\Pi^{\rho_{mix}} + \epsilon_\Pi^{\rho_{mix}}\sqrt{\frac{C_0}{N+M}}\right)\right\} + \sqrt{\frac{C}{N}}. \qquad (101)$$

*where $C_B = \left\| \frac{d_\mu^{\pi_E}}{d_\mu^{\pi_B}} \right\|_\infty$, $H$ is the horizon, $N$ is the number of expert state-action pairs, $M$ is the number of offline state-action pairs, $C_0 = 2\ln\frac{|\Pi||\mathcal{R}|}{\delta}$, and $C = \ln\frac{2|\mathcal{R}|}{\delta}$.*

*Proof.* By Lemma H.2, we have

$$\hat{L}(\pi_i, r_i) \leq L(\pi_i, r_i) + \sqrt{\frac{C}{N}} \qquad (102)$$

$$= \mathbb{E}_{(s,a)\sim d_\mu^{\pi_E}}\left[r_i(s,a)\right] - \mathbb{E}_{(s,a)\sim d_\mu^{\pi_i}}\left[r_i(s,a)\right] + \sqrt{\frac{C}{N}} \qquad (103)$$

$$= \frac{1}{H}\left(V_{r_i}^{\pi_E} - V_{r_i}^{\pi_i}\right) + \sqrt{\frac{C}{N}} \qquad (104)$$

$$= \frac{1}{H}\left(\sum_{h=1}^{H}\mathbb{E}_{(s_h,a_h)\sim d_h^{\pi_E}}A_{r_i,h}^{\pi_i}(s_h,a_h)\right) + \sqrt{\frac{C}{N}} \qquad (105)$$

$$\leq \frac{1}{H}\left(\sum_{h=1}^{H}\mathbb{E}_{s_h\sim d_h^{\pi_E}}\max_{a\in\mathcal{A}}A_{r_i,h}^{\pi_i}(s_h,a)\right) + \sqrt{\frac{C}{N}} \qquad (106)$$

$$= \frac{1}{H}\left(H\mathbb{E}_{s\sim d^{\pi_E}}\max_{a\in\mathcal{A}}A_{r_i}^{\pi_i}(s,a)\right) + \sqrt{\frac{C}{N}} \qquad (107)$$

where $C = \ln\frac{2|\mathcal{R}|}{\delta}$. The second line holds by the definition of $L(\pi_i, r_i)$, and the third line holds by the definition of the reward-indexed value function. The fourth line holds by the Performance Difference Lemma (PDL). Applying Lemma H.3, we have

$$\hat{L}(\pi_i, r_i) \leq \min\left\{ \epsilon_\Pi^{\rho_{mix}} + \epsilon_\Pi^{\rho_{mix}}\sqrt{\frac{C_0}{N}}, C_B\left(\epsilon_\Pi^{\rho_{mix}} + \epsilon_\Pi^{\rho_{mix}}\sqrt{\frac{C_0}{N+M}}\right)\right\} + \sqrt{\frac{C}{N}}. \qquad (108)$$

where $C_B = \left\| \frac{d_\mu^{\pi_E}}{d_\mu^{\pi_B}} \right\|_\infty$, $H$ is the horizon, $N$ is the number of expert state-action pairs, $M$ is the number of offline state-action pairs, $C_0 = 2\ln\frac{|\Pi||\mathcal{R}|}{\delta}$, and $C = \ln\frac{2|\mathcal{R}|}{\delta}$. $\qquad\square$

## H.2 FINITE SAMPLE ANALYSIS OF ALGORITHM 2

**Theorem H.5** (Sample Complexity of Algorithm 2). *Suppose that PSDP's accuracy parameter is set to $\epsilon = 0$. Then, upon termination of Algorithm 2, with probability at least $1 - \delta$, we have*

$$V^{\pi_E} - V^{\overline{\pi}} \le \underbrace{H \min \left\{ \epsilon_\Pi^{\rho_{mix}} + \epsilon_\Pi^{\rho_{mix}} \sqrt{\frac{C_{\Pi,\mathcal{R}}}{N}}, C_B \left( \epsilon_\Pi^{\rho_{mix}} + \epsilon_\Pi^{\rho_{mix}} \sqrt{\frac{C_{\Pi,\mathcal{R}}}{N+M}} \right) \right\}}_{\textit{Misspecification Error}} + \underbrace{H \sqrt{\frac{C_\mathcal{R}}{N}}}_{\textit{Statistical Error}} + \underbrace{H \sqrt{\frac{\ln |\mathcal{R}|}{n}}}_{\textit{Reward Regret}}$$

(109)

*where $H$ is the horizon, $N$ is the number of expert state-action pairs, $M$ is the number of offline state-action pairs, $n$ is the number of reward updates, $C_{\Pi,\mathcal{R}} = \ln \frac{|\Pi||\mathcal{R}|}{\delta}$, $C_\mathcal{R} = \ln \frac{|\mathcal{R}|}{\delta}$, and $1 \le C_B := \left\| \frac{d_\mu^{\pi_E}}{d_\mu^{\pi_B}} \right\|_\infty \le \infty$*

Theorem H.5 upper bounds the sample complexity of Algorithm 2 in the offline data setting. The bound differs from Theorem 3.3 in the following ways. First, the policy optimization error term vanishes by the assumption that $\epsilon = 0$. Importantly, the assumption of $\epsilon = 0$ is not necessary but rather convenient, as the $\epsilon > 0$ case was presented in Theorem 3.3. Second, offline data is incorporated into the reset distribution, resulting in a modified misspecification error. Finally, the finite expert sample regime is considered, resulting in statistical error of estimating the expert policy's state distribution $d_\mu^{\pi_E}$ with the distribution over samples $\rho_E$. We use this concentrability coefficient $C_B$ with respect to the expert policy for Theorem H.5 in order to obtain a tight bound on the sample complexity.

*Proof.* We consider the imitation gap of the expert and the averaged learned policies, $\overline{\pi}$,

$$V^{\pi_E} - V^{\overline{\pi}} = \frac{1}{n} \sum_{i=0}^n \left( \mathbb{E}_{\zeta \sim \pi_E} \left[ \sum_{h=1}^H r^*(s_h, a_h) \right] - \mathbb{E}_{\zeta \sim \pi_i} \left[ \sum_{h=1}^H r^*(s_h, a_h) \right] \right) \tag{110}$$

$$= \frac{1}{n} H \sum_{i=0}^n \left( \mathbb{E}_{(s,a) \sim d_\mu^{\pi_E}} [r^*(s,a)] - \mathbb{E}_{(s,a) \sim d_\mu^{\pi_i}} [r^*(s,a)] \right) \tag{111}$$

$$= \frac{1}{n} H \sum_{i=0}^n L(\pi_i, r^*) \tag{112}$$

$$\le \frac{1}{n} H \max_{r \in \mathcal{R}} \sum_{i=0}^n L(\pi_i, r) \tag{113}$$

where $n$ is the number of updates to the reward function. The second line holds by definition of $d_\mu^\pi$. The third line holds by definition of $L$. Applying the Statistical Difference of Losses (Lemma H.2), we have

$$V^{\pi_E} - V^{\overline{\pi}} \le \frac{1}{n} H \max_{r \in \mathcal{R}} \sum_{i=0}^n \left( \hat{L}(\pi_i, r) + \sqrt{\frac{C}{N}} \right) \tag{114}$$

$$= \frac{1}{n} H \max_{r \in \mathcal{R}} \sum_{i=0}^n \left( \hat{L}(\pi_i, r) - \hat{L}(\pi_i, r_i) + \hat{L}(\pi_i, r_i) + \sqrt{\frac{C}{N}} \right) \tag{115}$$

where $C = \ln \frac{2|\mathcal{R}|}{\delta}$ and $M$ is the number of state-action pairs from the expert. Applying the Reward Regret Bound (Lemma H.1), we have

$$V^{\pi_E} - V^{\overline{\pi}} \le \frac{1}{n} H \sum_{i=0}^n \left( \hat{L}(\pi_i, r_i) + \sqrt{\frac{C}{N}} \right) + H \sqrt{\frac{C_1}{n}} \tag{116}$$

where $C_1 = 2 \ln |\mathcal{R}|$. Applying the Loss Bound (Lemma H.4), we have

$$V^{\pi_E} - V^{\overline{\pi}} \le \frac{1}{n} H \sum_{i=0}^n \left( \min \left\{ \epsilon_\Pi^{\rho_{mix}} + \epsilon_\Pi^{\rho_{mix}} \sqrt{\frac{C_0}{N}}, C_B \left( \epsilon_\Pi^{\rho_{mix}} + \epsilon_\Pi^{\rho_{mix}} \sqrt{\frac{C_0}{N+M}} \right) \right\} \right.$$
$$\left. + \sqrt{\frac{C}{N}} \right) + H \sqrt{\frac{C_1}{n}}, \tag{117}$$

which simplifies to

$$V^{\pi_E} - V^{\overline{\pi}} \leq H \min \left\{ \epsilon_{\Pi}^{\rho_{\mathrm{mix}}} + \epsilon_{\Pi}^{\rho_{\mathrm{mix}}} \sqrt{\frac{C_0}{N}}, C_B \left( \epsilon_{\Pi}^{\rho_{\mathrm{mix}}} + \epsilon_{\Pi}^{\rho_{\mathrm{mix}}} \sqrt{\frac{C_0}{N+M}} \right) \right\} + H\sqrt{\frac{C}{N}}, + H\sqrt{\frac{C_1}{n}}, \tag{118}$$

where $C_B = \left\| \frac{d_\mu^{\pi_E}}{d_\mu^{\pi_B}} \right\|_\infty$, $H$ is the horizon, $N$ is the number of expert state-action pairs, $M$ is the number of offline state-action pairs, $n$ is the number of reward updates, $C_0 = 2\ln\frac{|\Pi||\mathcal{R}|}{\delta}$, $C = \ln\frac{2|\mathcal{R}|}{\delta}$, and $C_1 = 2\ln|\mathcal{R}|$. $\qquad\square$

**Proof Sketch of Corollary 4.1.** The proof follows from Theorem H.5, and more specifically, Lemma H.3. The notable change is, instead of using the concentrability coefficient with the expert policy, $\pi_E$, we use a concentrability coefficient with respect to the optimal realizable policy, $\pi^\star$, such that

$$C_B' := \left\| \frac{d_\mu^{\pi^\star}}{d_\mu^{\pi_B}} \right\|_\infty \tag{119}$$

Then, in Case 2 of Lemma H.3 (Equation 94), we perform the change of distribution from $d_\mu^{\pi_E}$ to $d_\mu^{\pi^\star}$ and then to $d_\mu^{\pi_B}$, resulting in an additional TV distance to bound the first distribution change. Notably, this factor is independent of our offline data, and because TV distance is positive, it can be added to both Case 1 and Case 2, resulting in a tradeoff that depends solely on the offline data distribution's coverage of $\pi^\star$'s state distribution, as well as the other standard terms ($\epsilon_\Pi$, number of expert samples, number of offline data samples, etc). We use the standard concentrability coefficient $C_B$ with respect to the expert policy for Theorem H.5 in order to obtain a tighter bound on the sample complexity.

# I  IMPLEMENTATION DETAILS

We describe the implementation details in this section. We compare GUITAR against two behavioral cloning baselines (Pomerleau, 1988) and two IRL baselines (Swamy et al., 2023). The first behavioral cloning baseline is trained exclusively on the expert data, $BC(\pi_E)$, and the second is trained on the combination of expert and offline data, $BC(\pi_E + \pi_b)$, when offline data is available. We compare against two IRL algorithms: (1) Swamy et al. (2021a)'s moment-matching algorithm, MM, a traditional IRL algorithm with the Jensen-Shannon divergence replaced by an integral probability metric, and (2) Swamy et al. (2023)'s efficient IRL algorithm, FILTER, that exclusively leverages expert data for resets.

We train all IRL algorithms with the same expert data and hyperparameters. The difference between MM, FILTER, and GUITAR can be summarized by the reset distribution they use. MM resets the learner to the true starting state (i.e. $\rho = \mu$), while FILTER resets the learner to expert states (i.e. $\rho = \rho_E$). GUITAR resets the learner to offline data (i.e. $\rho = \rho_B$), when offline data is available, and otherwise resets to a subset of the expert data. In other words, we isolate the effects of the reset distribution by using the same underlying IRL algorithm but simply change the reset distribution used in the RL optimizer step. This makes GUITAR easy to implement and easily adaptable to other IRL algorithms.

We adapt Ren et al. (2024)'s codebase for our implementation and follow their implementation details. We restate Ren et al. (2024)'s details here, with modifications where necessary. We apply Optimistic Adam (Daskalakis et al., 2017) for all policy and discriminator optimization. We also apply gradient penalties (Gulrajani et al., 2017) on all algorithms to stabilize the discriminator training. The policies, value functions, and discriminators are all 2-layer ReLu networks with a hidden size of 256. We sample 4 trajectories to use in the discriminator update at the end of each outer-loop iteration, and a batch size of 4096. In all IRL variants (MM, FILTER, and GUITAR), we re-label the data with the current reward function during policy improvement, rather than keeping the labels that were set when the data was added to the replay buffer. Ren et al. (2024) empirically observed such re-labeling to improve performance. We release a forked version of Ren et al. (2024)'s code: https://nico-espinosadice.github.io/efficient-IRL/.

We calculate the inter-quartile mean (IQM) and standard errors across seeds for all experiments.

## I.1  MUJOCO TASKS

### I.1.1  SETTING WITHOUT GENERATIVE MODEL ACCESS

We detail below the specific implementations used in all MuJoCo experiments (Ant, Hopper, and Humanoid).

| PARAMETER | VALUE |
|---|---|
| BUFFER SIZE | 1E6 |
| BATCH SIZE | 256 |
| $\gamma$ | 0.98 |
| $\tau$ | 0.02 |
| TRAINING FREQ. | 64 |
| GRADIENT STEPS | 64 |
| LEARNING RATE | LIN. SCHED. 7.3E-4 |
| POLICY ARCHITECTURE | 256 X 2 |
| STATE-DEPENDENT EXPLORATION | TRUE |
| TRAINING TIMESTEPS | 1E6 |

Table 1: Hyperparameters for baselines using SAC.

**Expert Data.**  To experiment under the conditions of limited expert data, we set the amount of expert data to be the lowest amount that still enabled the baseline IRL algorithms to learn. For Ant, this was 50 expert state-action pairs. For Humanoid, this was 100 expert state-action pairs. For Hopper, this was 600 expert state-action pairs.

| Joint | Default | Modified |
|---|---|---|
| Right hip_x | $[-25°, 5°]$ | $[0°, 30°]$ |
| Left hip_x | $[-25°, 5°]$ | $[0°, 30°]$ |
| Left knee | $[-160°, -2°]$ | $[-100°, 50°]$ |

(a) Joint angle range differences

| Body Part | Default | Modified | |
|---|---|---|---|
| Right thigh | 0.06 | 0.09 | (+50%) |
| Right foot | 0.075 | 0.1 | (+33%) |
| Left lower arm | 0.031 | 0.04 | (+29%) |

(b) Size parameter differences

Table 2: Comparison between default and modified Humanoid configurations.

**Offline Data.**   We generate the offline data by rolling out the expert policy with a probability $p_{\text{tremble}}^{\pi_B}$ of sampling a random action. $p_{\text{tremble}}^{\pi_B} = 0.25$ for the Ant environment and $p_{\text{tremble}}^{\pi_B} = 0.05$ for the Hopper and Humanoid environments.

**Discriminator.**   For our discriminator, we start with a learning rate of $8\mathrm{e}{-4}$ and decay it linearly over outer-loop iterations. We update the discriminator every 10,000 actor steps.

**Baselines.**   We train all behavioral cloning baselines for 300k steps for Ant, Hopper, and Humanoid. For MM and FILTER baselines, we follow the exact hyperparameters in Ren et al. (2024), with a notable modification to how resets are performed, discussed below. We use the Soft Actor Critic (Haarnoja et al., 2018) implementation provided by Raffin et al. (2021) with the hyperparameters in Table 1.

**Reset Substitute.**   We mimic resets by training a BC policy on the reset distribution specified by each algorithm. MM does not employ resets. FILTER's reset distribution is the expert data. GUITAR's reset distribution is a mixture of the expert and offline data. The BC roll-in logic follows Ren et al. (2024)'s reset logic. The probability of performing a non-starting-state reset (i.e. an expert reset in FILTER) is $\alpha$. If a non-starting-state reset is performed, we sample a random timestep $t$ between 0 and the horizon, and we roll out the BC policy in the environment for $t$ steps.

**GUITAR.**   GUITAR follows the same implementation and reset logic as FILTER, with the only change being the training data for the BC roll-in policy.

### I.1.2   MISSPECIFIED SETTING II: DIFFERENT DYNAMICS

For these experiments, we use the same parameters as in Section I.1.1, with the following changes.

**Expert Data.**   We use the same expert policy as in Section I.1.1. However, we do not consider the finite expert sample regime, so we use 100,000 state-action pairs from the expert policy. Notably, this expert policy uses the default configuration (i.e. the default MJCF file, MuJoCo's XML file specifying the robot's morphology, including link lengths and joint ranges) and we collect expert data by rolling out the policy with the default morphology.

**Offline Data.**   We generate the offline data by training a policy under a new configuration, specified below, otherwise following a similar setup to training the expert policy. We collect 100,000 state-action pairs from the offline policy by rolling it out with the new morphology.

**Baselines.**   We train all behavioral cloning baselines for 300k steps for Humanoid and Ant. For MM and FILTER baselines, we otherwise follow the exact hyperparameters in Ren et al. (2024), except for varying the robot's morphology. We use the Soft Actor Critic (Haarnoja et al., 2018) implementation provided by Raffin et al. (2021) with the hyperparameters in Table 1.

**Resets.**   MM resets the learner to the true starting state, so no changes are needed. FILTER resets the learner to expert states. However, due to the difference in morphology between the expert and learner, it is not possible to reset the learner to expert states. Therefore, FILTER cannot be applied in this misspecified setting.

| Joint | Default | Modified |
|---|---|---|
| All Hip joints | $[-30°, 30°]$ | $[-20°, 40°]$ |
| Front Ankles (1,4) | $[30°, 70°]$ | $[40°, 80°]$ |
| Back Ankles (2,3) | $[-70°, -30°]$ | $[-60°, -20°]$ |

(a) Joint angle range differences

| Body Part | Default | Modified | |
|---|---|---|---|
| Auxiliary segments | 0.08 | 0.11 | (+38%) |
| Leg segments | 0.08 | 0.11 | (+38%) |
| Ankle segments | 0.08 | 0.11 | (+38%) |

(b) Geometry size differences

Table 3: Comparison between default and modified Ant configurations. All changes are applied consistently to all four legs.

**GUITAR.** GUITAR follows the same implementation as FILTER, with the notable change of its reset distribution. Unlike FILTER, GUITAR resets the learner to the offline data, and in contrast to resetting to expert states, is possible in this case, since the offline behavior policy and the learner have the same morphology.

**Learner.** The learner, in both the training and evaluation environments, has a modified morphology, as indicated by Table 2 and 3 for Humanoid and Ant, respectively.

## I.2 D4RL TASKS

### I.2.1 RESETTING TO SUBSETS OF $\pi^\star$'S STATE DISTRIBUTION

For the two Antmaze-Large tasks, we use the data provided by Fu et al. (2020) as the expert demonstrations. We append goal information to the observation for all algorithms following Ren et al. (2024); Swamy et al. (2023). For our policy optimizer in every algorithm, we build upon the TD3+BC implementation of Fujimoto & Gu (2021) with the default hyperparameters.

**Expert Data and Discriminator.** We use the relevant D4RL dataset to learn the discriminator. For our discriminator, we start with a learning rate of $8e - 3$ and decay it linearly over outer-loop iterations. We update the discriminator every 5,000 actor steps.

**Baselines.** For behavioral cloning, we run the TD3+BC optimizer for 500,000 steps while zeroing out the component of the actor update that depends on rewards. We use a reset proportion of $\alpha = 1.0$. We provide all runs with the same expert data. All IRL algorithms are pretrained with 10,000 steps of behavioral cloning on the expert dataset.

**Reset Distributions.** We reset GUITAR to various reset distributions. We consider the expert dataset, short trajectories (of length less than 500) in the expert dataset, and long trajectories (of length greater than or equal to 500) in the expert dataset. We use each of these datasets as different reset distributions. For the D4RL tasks, we perform true state resets (i.e. reset the learner to states in the reset distribution), rather than perform BC roll-ins as done in the MuJoCo tasks. Notably, IRL with resets to the true starting state distribution (i.e. no selective reset distribution) has been well studied by prior work (Swamy et al., 2023; Ren et al., 2024) and observed to not solve the Antmaze-Large tasks.

### I.2.2 MISSPECIFIED SETTING I: UNREACHABLE EXPERT STATES - BLOCK OBSTRUCTION

For the expert action misspecification experiments, we train an RL expert using TD3+BC. We append goal information to the observation for all algorithms following Ren et al. (2024); Swamy et al. (2023). For our policy optimizer in every algorithm, we build upon the TD3+BC implementation of Fujimoto & Gu (2021) with the default hyperparameters.

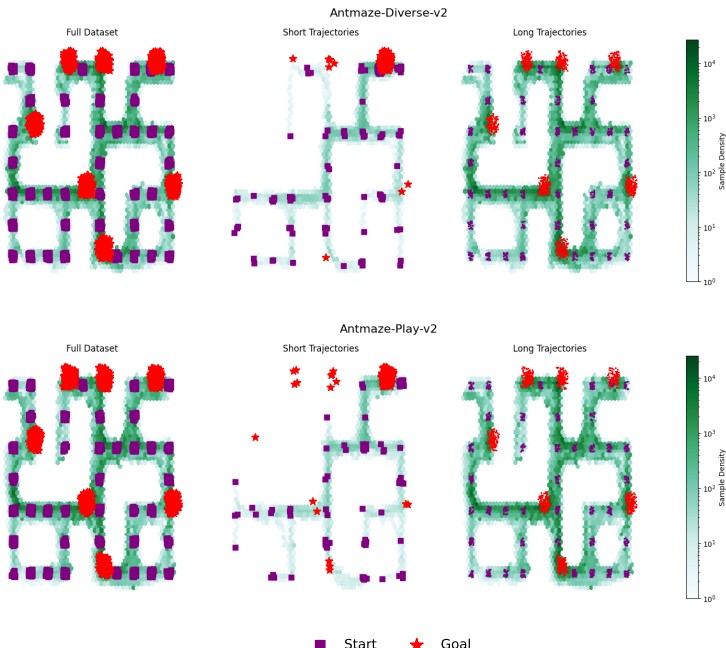

Figure 5: We plot the coverage of the D4RL `Antmaze-Large` expert data, including the full dataset, short trajectories (episodes shorter than 500 steps), and long trajectories (episodes 500 steps or longer).

**Expert Data and Offline Data.** We collect expert data by first training an RL expert using TD3+BC on the following maze map:

$$
M_{\pi_E} := \begin{bmatrix} 1 & 1 & 1 & 1 & 1 \\ 1 & \mathbf{0} & 0 & 0 & 1 \\ 1 & \mathbf{R} & 1 & \mathbf{G} & 1 \\ 1 & \mathbf{1} & 0 & 0 & 1 \\ 1 & 1 & 1 & 1 & 1 \end{bmatrix}
$$

where **R** is the starting state and **G** is the goal state. The expert policy is then rolled out for 100,000 state-action samples. We collect the offline data by training an RL agent using TD3+BC on the following maze map:

$$
M_{\pi_B} := \begin{bmatrix} 1 & 1 & 1 & 1 & 1 \\ 1 & \mathbf{1} & 0 & 0 & 1 \\ 1 & \mathbf{R} & 1 & \mathbf{G} & 1 \\ 1 & \mathbf{0} & 0 & 0 & 1 \\ 1 & 1 & 1 & 1 & 1 \end{bmatrix}
$$

and then rolling out the policy for 100,000 state-action samples. The differences between $M_{\pi_E}$ and $M_{\pi_B}$ are bolded.

**Discriminator.** We use the expert data to learn the discriminator. For our discriminator, we start with a learning rate of $8e-3$ and decay it linearly over outer-loop iterations. We update the discriminator every 5,000 actor steps. We use 10 sample trajectories for the discriminator update. Since this is a strongly misspecified setting, we only use the $x$-position of the agent as the input to the discriminator for all IRL algorithms. The explanation for this design choice is that the learner's and the expert's behaviors may differ how they solve the maze (i.e. the $y$-position), but the goal is to finish the maze in some way (i.e. dependent on the $x$-position).

**Baselines and Reset Distributions.** For behavioral cloning, we run the TD3+BC optimizer for 500,000 steps while zeroing out the component of the actor update that depends on rewards. We use a reset proportion of $\alpha = 1.0$. We provide all runs with the same expert data. Due to the strong misspecification in this task, we do not pretrain the IRL algorithms with behavioral cloning.

`MM` is reset to the true starting state, while `FILTER` is reset to the expert data. `GUITAR` is reset to the offline data (i.e. $\pi_B$'s data).

**Misspecification.** The learner is trained and evaluated on the map $M_{\pi_B}$. Notably, this difference ensures that the expert policy solves the maze via one path, and the learner must solve it in a different path, where the only shared states can be the start and goal states. For the expert and offline data collection, as well as the learner's environment, we add a `BackWallGoalWrapper`, which allows the expert to achieve a reward of 1.0 by solving the maze through the "top hallway," but due to the block in the learner's maze, the learner is only able to achieve a reward of 0.9 by solving the maze through the "bottom hallway." The `BackWallGoalWrapper`'s required $x$-position to solve the maze is 4.0.

It should be noted that this is not the typical Antmaze-Umaze task, which can typically be solved by IRL approaches. In this setting, we consider the strongly misspecified setting, where the expert policy solves the maze one way, but the learner must solve it differently due to differences in the maze at "test-time." The degree of exploration difficulty due to the misspecification is likely the reason for the poor performance of the baselines.

### I.2.3 MISSPECIFIED SETTING I: UNREACHABLE EXPERT STATES – TIME CONSTRAINT

**Expert Data and Offline Data.** We train an expert policy via RL to solve the `Antmaze-Umaze-v2` task and roll out the policy in the default Umaze configuration to collect 100,000 state-action pairs. We use the same policy for the offline data, but we stop the roll-out once the "safety constraint" is reached (described below). This means that the offline data consists of only states from the "first hallway" in the maze.

**Misspecification.** To incorporate a "safety constraint," such that the learner is not able to reach the "bottom halllway" of the maze, we impose a time constraint in the learner's environment, such that after 'T=50' steps, the episode terminates.

**Discriminator.** We use the expert data to learn the discriminator. For our discriminator, we start with a learning rate of $8e - 3$ and decay it linearly over outer-loop iterations. We update the discriminator every 5,000 actor steps. We use 10 sample trajectories for the discriminator update. We use full agent's observation for the discriminator input.

**Baselines and Reset Distributions.** For behavioral cloning, we run the TD3+BC optimizer for 500,000 steps while zeroing out the component of the actor update that depends on rewards. We use a reset proportion of $\alpha = 1.0$. We provide all runs with the same expert data. Due to the strong misspecification in this task, we do not pretrain the IRL algorithms with behavioral cloning.

`MM` is reset to the true starting state, while `FILTER` is reset to the expert data. `GUITAR` is reset to the offline data (i.e. $\pi_B$'s data).

## J  SETTING WITHOUT GENERATIVE MODEL ACCESS

We consider two additional constraints common in real-world robotics applications: settings with finite expert data and environments without generative model access (i.e. where the robot cannot be reset to an arbitrary state). To ensure we train in the low-data regime, we used the minimum amount of expert data that allowed the baseline IRL algorithm (MM) to learn in each environment (notably, less than one full episode). The offline data was generated by rolling out the pretrained expert policy with a probability $p_{\text{tremble}}^{\pi_b}$ of sampling a random action. We mimic resets by rolling in with a BC policy trained on the corresponding reset distribution. More specifically, FILTER's reset distribution consists of the the expert states, so FILTER rolls in with $\text{BC}(\pi_E)$. GUITAR's reset distribution is a mixture of expert and offline states, so GUITAR rolls in with $\text{BC}(\pi_E + \pi_b)$. MM continues to reset to the environment's true starting state.

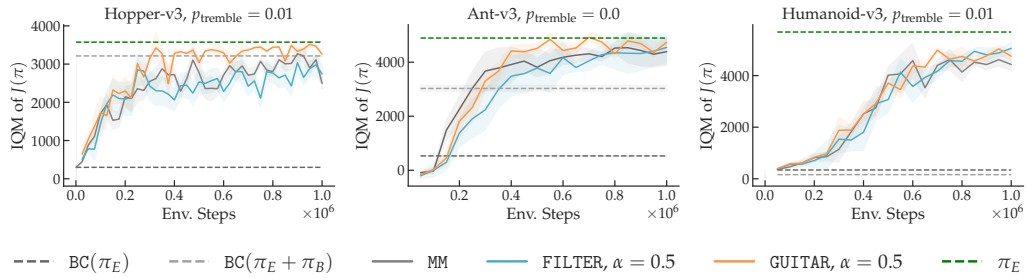

Figure 6: **Environment without arbitrary reset access.** Standard errors are computed across 5 seeds. Expert data is a partial trajectory (i.e. a subset of one full trajectory).

Based on Theorem 4.1, the sample efficiency of IRL should improve as the reset distribution's coverage of the expert's state distribution improves. More formally, this is when

$$C_B = \left\| \frac{d_\mu^{\pi_E}}{\rho} \right\|_\infty \to 1, \tag{120}$$

where $d_\mu^{\pi_E}$ is the expert's state distribution and $\rho$ is the reset distribution. Since GUITAR resets to $\text{BC}(\pi_E + \pi_b)$, the performance of $\text{BC}(\pi_E + \pi_b)$ is an estimation of the GUITAR's reset distribution's coverage of the expert's state distribution, and correspondingly for FILTER's reset distribution and $\text{BC}(\pi_E)$. From Figure 6, we see that as the reset distribution's coverage of the expert's states improves—as measured by the corresponding BC performance—so does the performance of the IRL algorithm. In the case where the BC performance is poor (Humanoid-v3), there is no observable benefit to modified resets.

# K    USEFUL LEMMAS

**Theorem K.1** (Hoeffding's Inequality). *If $Z_1, \ldots, Z_M$ are independent with $P(a \leq Z_i \leq b) = 1$ and common mean $\mu$, then, with probability at least $1 - \delta$,*

$$|\overline{Z}_M - \mu| \leq \sqrt{\frac{c}{2M} \ln \frac{2}{\delta}} \tag{121}$$

*where $c = \frac{1}{M} \sum_{i=1}^{M} (b_i - a_i)^2$.*

**Lemma K.2** (Online Mirror Descent Regret). *Regret is defined as*

$$\lambda_N = \frac{1}{N} \sum_{t=1}^{N} \ell(\hat{\mathbf{y}}_t, z_t) - \inf_{\mathbf{f} \in \mathcal{F}} \frac{1}{N} \sum_{t=1}^{N} \ell(\mathbf{f}, z_t). \tag{122}$$

*Given $\mathcal{F} = \Delta(\mathcal{F}')$ and $\langle \mathbf{f}, \nabla_t \rangle = \mathbb{E}_{f' \sim \mathbf{f}}[\ell(f', (x_t, y_t))]$, where $\sup_{\nabla \in \mathcal{D}} \|\nabla\|_\infty \leq B$, let $R$ be any 1-strongly convex function. If we use the Mirror descent algorithm with $\eta = \sqrt{\frac{2 \sup_{\mathbf{f} \in \mathcal{F}} R(\mathbf{f})}{N B^2}}$, then,*

$$\lambda_n \leq \sqrt{\frac{2B^2 \sup_{\mathbf{f} \in \mathcal{F}} R(\mathbf{f})}{N}}. \tag{123}$$

*If $R$ is the negative entropy function, then $\sup_{\mathbf{f} \in \mathcal{F}} R(\mathbf{f}) \leq \log |\mathcal{F}'|$.*

