# OpenReview forum: "Efficient Imitation under Misspecification"
_ICLR.cc/2025/Conference — ICLR 2025 Poster_

### Official Review · Reviewer_iMEE · 2024-10-31

**Soundness:** 3
**Presentation:** 3
**Contribution:** 3
**Rating:** 8
**Confidence:** 3

**Summary:**

This paper considers the interactive imitation-learning setting where policy realizability is violated. The paper further considers the setting where the reset distribution is a mixture of expert and a suboptimal policy, a more practical setting than reset distribution induced by just the expert policy. Based on these assumptions, the paper provides theoretical results on when the former reset distribution is more beneficial. Finally, the paper conducts few experiments to validate the theoretical findings and also identify potential gaps for future analysis.

**Strengths:**

- The paper is well written and describes the theoretical results intuitively.
- The misspecification setting is definitely a setting where previous works did not consider.
- This work puts in a good effort to bridge the theory and application gap with experiments which I appreciate a lot

**Weaknesses:**

- Small writing nitpicks:
	- Page 6, equation 10 (or Algorithm 2): Should $n$ be $N$ (or the other way around) for the reward regret?
	- Page 6, line 315: Perhaps there should be some statement indicating why we cannot be more aggressive on $\epsilon$ (e.g. $\epsilon = 1/H^2$) (or why it does not make sense for this to happen).
- While I see some value of this theoretical work, I have a concern about how this paper is positioned which influence the score by a lot:
	- For a theory paper, I feel this result is not groundbreaking, in the sense that the bounds seems to be very standard. Perhaps the authors can clarify this and whether I am missing anything.
	- For a empirical paper, (1) I have trouble convincing myself why rolling-in with BC policy is a valid approach (assuming the BC policy was trained with the scarce data provided in the dataset), and (2) there can be more analysis on the experiments including analyzing the extent of misspecification and the potential gap on "trajectory length" the paper has raised.
- For experiments, an alternative to using $p_{tremble}$ is to have mixture like $\alpha \rho_E + (1 - \alpha) \mu$ which I believe can be a way more realistic setting than random actions during data collection.

**Questions:**

- To follow up on the previous point, I am curious how misspecification is validated in the experiments? Did the authors try running behavioural cloning on a large dataset and find that there is an inherit performance gap?
- On page 10, lines 515-516: "When we improve the roll-in distribution by adding offline data, we see..." I am having trouble relating this result with Figure 2, can the paper please clarify this statement?
	- Perhaps I am confused about whether the roll-in for `FILTER` is using $\pi_E$ rather than $BC(\pi_E)$?

---

> ### Author Response · Authors · 2024-11-24
> **Rebuttal**
>
> Thank you for your review. We address the concerns raised.
>
> ## 1. Theoretical work and concerns
> > While I see some value of this theoretical work, I have a concern about how this paper is positioned which influence the score by a lot:
>
> > For a theory paper, I feel this result is not groundbreaking, in the sense that the bounds seems to be very standard. Perhaps the authors can clarify this and whether I am missing anything.
>
> As detailed in the Introduction, the realizability assumption is quite strong and often untrue in robotics and self-driving applications due to the expert having privileged information over the expert (Kumar et al., 2021; 2022; Liang et al., 2024; Swamy et al., 2022b) or morphological differences between robots and humans (Zhang et al., 2024; He et al., 2024; Al-Hafez et al., 2023). However, prior work in efficient interactive imitation learning has not considered the misspecified setting (i.e. where the expert policy is not realizable by the learner’s policy class). **It is an open question whether interactive imitation learning methods can efficiently avoid quadratically compounding errors.**
>
> **We present the first bounds on efficient interactive imitation *in the misspecified setting*.** While the bounds themselves are not necessarily tighter than prior work, the setting we consider is novel – and notably does not rely on the unrealistic realizability assumption. Prior work assumes that since $\pi_E \in \Pi$, then after enough optimization (i.e. policy search), the expert policy can be recovered. We conduct a more careful finite sample analysis and decompose the error into distinct components, including optimization error, misspecification error, and finite sample error. Additionally, we do not need a queryable expert like DAgger (Ross et al., 2011), nor global search like IRL (Ziebart et al., 2008).
>
> Additionally, **we prove how offline data can improve the sample efficiency in interactive imitation algorithms, *without relying on strong assumptions about the offline data’s structure*.**
>
> ## 2. Empirical work and concerns
> > For a empirical paper,
> > 1. I have trouble convincing myself why rolling-in with BC policy is a valid approach (assuming the BC policy was trained with the scarce data provided in the dataset), and
> > 2. there can be more analysis on the experiments including analyzing the extent of misspecification and the potential gap on "trajectory length" the paper has raised.
>
> > To follow up on the previous point, I am curious how misspecification is validated in the experiments? Did the authors try running behavioral cloning on a large dataset and find that there is an inherit performance gap?
>
> *Response to (2)*: In our revised paper, **we add a new experiment with far greater misspecification**, under which we see a considerable gap between our method (GUITAR), and the interactive baselines (MM and FILTER) and offline baseline (BC). In our new setting, the expert solves the maze by traversing one way through the maze. The learner, however, is unable to take the actions necessary to follow the expert’s course through the maze. (We restrict the learner’s movement by adding additional blocks in the path of the expert.) **Therefore, the learner must solve the maze through a *different route* through the maze.** This experiment is performed with true resets.
>
> *Response to (1)*: We also consider the setting without access to true resets. (For example, in real robots, resetting the robot to an arbitrary state is often impossible.) We propose an efficient, alternative approach to performing true resets that effectively resets the learner to the BC policy’s distribution (i.e. by rolling in with the BC policy). This method also gives us an exact way to measure the quality of the reset distribution. We clarified this in the Experiments section.
>
> ### Question on roll-in distributions
> > On page 10, lines 515-516: "When we improve the roll-in distribution by adding offline data, we see..." I am having trouble relating this result with Figure 2, can the paper please clarify this statement?
>
> We added further discussion and clarification of reset distributions in Sections 4 and 6. In Figure 2, FILTER's reset distribution is $\text{BC}(\pi_E)$, while GUITAR's reset distribution is $\text{BC}(\pi_E + \pi_B)$. This is accomplished in FILTER and GUITAR by rolling in with the corresponding BC policy. In the Hopper environment in Figure 2, we observe a difference between the performance of $\text{BC}(\pi_E + \pi_B)$ and $\text{BC}(\pi_E)$, which indicates a difference in the quality of reset (i.e. roll-in) distributions between GUITAR and FILTER. Reflecting the trend in reset distribution quality, we observe a performance difference between GUITAR and FILTER. In contrast, in the Walker environment, $\text{BC}(\pi_E + \pi_B)$ and $\text{BC}(\pi_E)$ perform similarly, indicating a similar reset distribution quality. The trend is reflected in the similar performance between FILTER and GUITAR.

---

> > ### Comment · Reviewer_iMEE · 2024-11-25
> >
> > Thank you for the detailed response. I agree that the raised theoretical questions were open and this paper has addressed them, and further provide new findings on the scenario with less-optimal offline data. I also thank the authors for clarifying the experimental setup and conducting extra experiments to further demonstrate the misspecification aspect. I've increased my score accordingly.

---

### Official Review · Reviewer_qn2v · 2024-11-03

**Soundness:** 3
**Presentation:** 3
**Contribution:** 2
**Rating:** 6
**Confidence:** 4

**Summary:**

The paper studies inverse reinforcement learning (IRL) under the misspecification setup, i.e., the experts policy is not realizable within the policy class used for learning the policy. The paper presents a theoretical analysis of this case and introduces the measure of Reward-Agnostic Policy Completeness Error to quantify the misspecification under the imitation learning / IRL setup. This measure is used to obtain sample complexity bounds under the misspecification assumptions. Moreover, the paper argues that existing IRL algorithms that do not consider the misspecification assumption, rely on the expert's distribution for resetting, which is now in the misspecication setup not ideal any more as the expert distribution might contain states from which the learned policy can not reach the goal. For this case, the paper studies using additional offline data for the resetting the environment and the theoretical conditions when this is beneficial. The paper also contains limited experimental evaluation of this algorithm.

**Strengths:**

- the paper formalizes a so far less considered problem, policy misspecification. The paper also introduces new interesting quantities such as the reward agnostic  Policy Completeness Error
- the theoretical analysis of the paper is very interesting and looks sound.
- the idea of using offline for the reset distribution is interesting

**Weaknesses:**

- the assumptions made in the paper for formalizing misspecification do not really fit the motivation introduced in the introduction. Here, misspecification is motivated by (i) "distinct perception" and (ii) "distinct action spaces" due to different morphology. I am unsure whether (i) is really covered by the given theory and algorithm as this would require to use the partial observable setup (for both, expert and policy) where the theory probably does not hold any more. At least implementing resets to offline states might be very hard in the partial observable setup. Concerning (ii), while this could be covered by the theory as far as I understand, the experiments do not cover such a setup at all. This should be clarified in the paper.
- misspecification is mainly considered if the policy  parametrization is not strong enough to take the optimal action in each state. However, as DNNs are very expressive representations, I am wondering how much this is really an issue in most setups (i.e. most DNN policies can reproduce the optimal policy, at least in the setups considered in the experiments of the paper).
- The experiments are rather limited and do not show any application under "misspecification", which is the main contribution of the paper. For all experiments, the experts policy can be realized by the learned policy, so I do not fully see the point if studying misspecifation in this setup. While (i) might be hard to realize, the authors could have presented experiments where (ii) is the case (see above). This can be done in gridworlds as well as for continuous control (i.e. the expert has other action limits then the behavior policy). Such experiments would be much better suited to illustrate the benefits of the approach.
- The presented experiments also hardly show any benefit of the presented method. E.g. I can not see any benefit in Figure 2. Alsothe experiment of Figure 3 is not well motivated. In this experiment, if I have the expert data, I can directly use it for the reset, why should I use less (I.e. the "short trajectories")? Seems to give the same performance...

**Questions:**

- Please clearly motivate in which setup of misspecification you oberate (different state observations or different actions)
- Provide a better motivation of the experiments and how they show a benefit in the misspecification setup
- Experiments with an actual misspecification between expert and learned policy (even if only gridworld) would really help the paper
- I was unclear about the mirror descent update rule for the reward. What the entropy of the reward R here? Is the rewward stochastic such that we can compute entropies? More information here would be useful (even though that might be described in previous papers, its not general knowledge and should be described in the paper as well briefly for self-consistency)
- I was also unclear why in the case of approximate policy completeness, we have O(1) for the sample complexity error bounds. Please provide more detail here.

In general I like the paper, but the experiments do not show any application where misspecifcation is an issue. This is the main motivation of the paper and hence this should also be part of the evaluation. Happy to raise my score if the authors can provide experimental insights there.

Post-Rebuttal: The new experiment show now a clear application in misspecification. Raised my score to 6.

---

> ### Author Response · Authors · 2024-11-25
> **Rebuttal**
>
> Thank you for your review. We believe we have addressed your concerns, especially regarding lack of misspecification in experiments. We discuss the new experiment we considered below.
>
> ## 1. Assumptions about misspecification
> > the assumptions made in the paper for formalizing misspecification do not really fit the motivation introduced in the introduction. Here, misspecification is motivated by (i) "distinct perception" and (ii) "distinct action spaces" due to different morphology...
>
> Our theory and algorithm can handle any source of misspecification, including both “distinct perception” and “distinct action spaces” highlighted above–we do not need a case-by-case analysis. (The approximate policy completeness structural condition works in both cases and is not reliant on fully observable information.) This is one of the strengths of our analysis. In terms of the second concern, we propose and validate one method for mimicking resets in the setting without access to resets (Section 6.2 of the revised paper). We address the third concern below.
>
> ## 2. Misspecification and DNNs
> > misspecification is mainly considered if the policy parametrization is not strong enough to take the optimal action in each state. However, as DNNs are very expressive representations, I am wondering how much this is really an issue in most setups...
>
> We agree that DNNs are expressive representations, but this is not the issue we are highlighting. The misspecification arises because (1) the robot itself cannot perform the expert’s actions/behavior, irrespective of the policy parameterization, due to morphological differences between robots and/or humans (Zhang et al., 2024; He et al., 2024; Al-Hafez et al., 2023). Additionally, the misspecification issue arises because (2) the expert has access to privileged information. No DNN can read a text-heavy road sign if it isn’t in the state/observation.
>
> ## 3. Limited misspecification in experiments
> > The experiments are rather limited and do not show any application under "misspecification", which is the main contribution of the paper. For all experiments, the experts policy can be realized by the learned policy, so I do not fully see the point if studying misspecifation in this setup. While (i) might be hard to realize, the authors could have presented experiments where (ii) is the case (see above). This can be done in gridworlds as well as for continuous control...
>
> The existing experiments considered misspecification by adding “trembling,” or some random action selection, for the learner, thus preventing the learner from exactly mimicking the expert policy. To address the critique that our experiments did not sufficiently consider the misspecified setting, **we introduce a new experiment with far larger misspecification.** In our new setting, the expert solves the maze by traversing one way through the maze. The learner, however, is unable to take the actions necessary to follow the expert’s course through the maze. (We restrict the learner’s movement by adding additional blocks in the path of the expert.) Therefore, **the learner must solve the maze through a *different route* through the maze.** This new experiment addresses the setting of strong misspecification that the reviewer has asked for. For the camera ready version, we will have more versions of similarly misspecified tasks.
>
> ## 4. Benefit of proposed method
> > The presented experiments also hardly show any benefit of the presented method. E.g. I can not see any benefit in Figure 2...
>
> The new experiment setting shows a significant difference between our algorithm and the baselines and thus highlights the value of our method in hard exploration and strongly misspecified tasks. Additionally, there was a mistake in our previous revision’s legend for one of the experiments. We further clarified that the motivation behind Figure 2’s experiments was to show a trend, specifically how the reset distribution’s quality affects the speedup seen in the IRL algorithm. We elaborated on this in Section 6.
>
> ## 5. Mirror descent update
> The negative entropy function is used in Online Mirror Descent (OMD) to project from the dual space to the primal space. We simplified the description of our GUITAR algorithm by highlighting that any no-regret algorithm will work; we simply choose Online Mirror Descent for its strong theoretical guarantees. We then present a more thorough discussion of OMD in Appendix D.
>
> ## 6. Approximate policy completeness
> > I was also unclear why in the case of approximate policy completeness, we have O(1) for the sample complexity error bounds.
> $O(1)$ is not the sample complexity of the error bounds. It is the complexity of the reward-agnostic policy completeness condition. We consider the structural condition of “approximate policy completeness,” which we define to be when $\epsilon_\Pi = O(1)$, meaning the reward-agnostic policy completeness term is not dependent on the horizon of the MDP.

---

> > ### Comment · Reviewer_qn2v · 2024-11-25
> > **Rebuttal Response**
> >
> > Many thanks for the exhaustive answer. The paper is now in a much better shape and I also like the newly added experiment. Given the nice theoretical contribution and the improved experiments, I will increase my rating to 6.

---

### Official Review · Reviewer_EKUW · 2024-11-04

**Soundness:** 2
**Presentation:** 2
**Contribution:** 2
**Rating:** 5
**Confidence:** 4

**Summary:**

The paper titled addresses a significant challenge in imitation learning (IL) by considering the setting when the expert's policy may not be fully realizable within the learner's policy class. This misspecified setting reflects real-world scenarios where the learner's capabilities may differ from the expert's, such as in robotic applications. It introduces reward-agnostic policy completeness to handle compounding errors even with this misspecification and proposes the GUITAR algorithm, which combines expert and offline data for efficient resets. Theoretical analysis and experiments on continuous control tasks are provided.

**Strengths:**

- The paper is theoretically robust, providing a thorough analysis of error sources in IL—optimization, finite sample, and misspecification errors—and offering a formal lower bound that emphasizes the challenges of efficient IRL under misspecification.

- By addressing both limited data and misspecification, the paper proposes practical improvements for settings where ideal conditions (perfect data, full realizability) are unattainable, a scenario commonly encountered in robotics and autonomous driving.

**Weaknesses:**

- While the use of offline data is well-motivated, additional analysis of how different qualities and types of offline data affect performance could strengthen the results and offer practical insights for selecting such data.

- The GUITAR algorithm, with its dependence on multiple reset distributions and dual optimization steps, may be challenging to implement and deploy in resource-constrained environments. Simplifications or guidelines on choosing reset distributions could make it more accessible.

- The authors' choice of not including the codes during the review process make the claims less transparent.

- The main results are only provided in three MuJoCo environments. However, as has been understood in literature, compounding errors for offline IL algorithms like BC happen when the environment is stochastic. But the MuJoCo environments are almost deterministic where BC seems to perform really well [1] with very limited data. I wonder why no stochastic environments are used for evaluation.

[1] Li, Ziniu, et al. "Rethinking ValueDice: Does it really improve performance?." arXiv preprint arXiv:2202.02468 (2022).

**Questions:**

- What is happening in the AntMaze experiment provided in the Appendix? I don't get what the conclusion from that experiment is.

- The algorithm's design is notably complex, involving a dual optimization loop with reset-based IRL, reward updates via mirror descent, and policy updates that mix offline and expert data resets. Could you please talk a little bit about the computational complexity of the method?

- Could you include a brief description of PSDP in the main paper to make it more self-contained?

---

> ### Author Response · Authors · 2024-11-24
> **Rebuttal**
>
> Thank you for your review. Below, we address the concerns raised, and have revised our paper accordingly.
>
> ## 1-2. Guitar algorithm design and implementation
> > The GUITAR algorithm, with its dependence on multiple reset distributions and dual optimization steps, may be challenging to implement and deploy in resource-constrained environments...
>
> > The algorithm's design is notably complex, involving a dual optimization loop with reset-based IRL, reward updates via mirror descent, and policy updates that mix offline and expert data resets. Could you please talk a little bit about the computational complexity of the method?
>
> **Our algorithm’s dual optimization loop is standard for inverse RL**. The algorithm we propose is conceptually simpler than standard inverse RL: we replace the global search of RL with the local search of “staying on the expert’s path.” A key strength of our algorithm, GUITAR, is that it is easy to implement in practice. GUITAR can be implemented via a straightforward environment reset wrapper, thus **retaining the level of computational complexity as the other IRL baselines** (MM and FILTER), while **improving on sample efficiency.**
>
> More generally, IRL algorithms are known to be more statistically efficient – with respect to expert data – than offline imitation algorithms (Swamy et al., 2021a). Our paper improves the sample efficiency – with respect to environment interactions – of IRL in the misspecified setting (Theorem 3.3). Our implementation of GUITAR does not impose any additional computation over the baseline IRL algorithms (Appendix G).
>
> ## 3. Analysis of different qualities and types of offline data
> > While the use of offline data is well-motivated, additional analysis of how different qualities and types of offline data affect performance could...
>
> Thank you for this suggestion. We updated our paper based on this feedback. In our revision, **we devote more discussion to the impact of reset distributions in the misspecified setting** – both from theory and empirical perspectives – in a way that we feel will give the reader clearer takeaways on choosing reset distributions in the misspecified setting.
>
> ## 4. Description of PSDP
> > Could you include a brief description of PSDP...
>
> We revised the section describing our algorithm. We emphasized that **our method can work with any RL algorithm**, including popular ones such as SAC, TRPO, and PPO. We specifically choose PSDP for its theoretical guarantees, and added greater discussion in Appendix D, and the full PSDP procedure is described in Algorithm 3.
>
> ## 5. Code
> > The authors' choice of not including the codes...
>
> We agree that releasing code is an important part of the process, and **we will release our code publicly with the release of the paper.**
>
> ## 6. Experiment environments
> > The main results are only provided in three MuJoCo environments. However... compounding errors for offline IL algorithms... happen when the environment is stochastic...
>
> We agree that one of the areas where BC fails is in stochastic environments. To incorporate greater stochasticity in the Mujoco environments, we add noise in the environment by randomly selecting an action with probability $p_{tremble}$. However, we wish to **emphasize that a primary cause of BC's failure is misspecification** (Ross et al., 2011), and ***the misspecified setting is the focus of our paper***. In our revision, **we added an experiment under which there is greater misspecification** than simply the random action $p_{tremble}$. In our new setting, the expert solves the maze by traversing one way through the maze. The learner, however, is unable to take the actions necessary to follow the expert’s course through the maze. (We restrict the learner’s movement by adding additional blocks in the path of the expert.) Therefore, **the learner must solve the maze through a *different route* through the maze**. Under such settings, we see a significant difference between our method (GUITAR) and the IRL baselines (MM and FILTER) and offline baselines (BC).
>
> ## 7. AntMaze experiment in Appendix
> > What is happening in the AntMaze experiment provided in the Appendix...
>
> **We added further explanation and a key to Figure 4 in Appendix H.** The figure is meant to demonstrate the state distribution of different components of the D4RL dataset, when parsed by short and long trajectories, showing where the ant is reset to, where it travels, and where the goals are in the dataset's episodes. The purpose of the associated experiment (Figure 5) is to empirically test the effectiveness of different reset distributions for ant maze. In other words, are there subsets of the expert’s data that are better to reset to than others? We find the answer is yes: resetting to “short” trajectories in the dataset is far more beneficial than resetting to longer trajectories, suggesting that resetting to states closer to the goal improves learning more than resetting to states farther from the goal.

---

> > ### Comment · Reviewer_EKUW · 2024-11-26
> >
> > - Dual Optimization was standard for inverse RL few years ago. Many works have appeared in inverse RL literature that considers only single optimization loop, for ex: see [1].
> >
> > - I would have liked to see the code as part of the supplementary materials for me to run it and verify the claims.
> >
> > Thank you for responding to my concerns. I would like to maintain my original score.
> >
> > [1] Garg, Divyansh, et al. "Iq-learn: Inverse soft-q learning for imitation." Advances in Neural Information Processing Systems 34 (2021): 4028-4039.

---

> ### Author Response · Authors · 2024-11-26
> **Response**
>
> Q functions depend on the MDP’s dynamics. It was shown in [2] that the learning Q function approach of **$\texttt{IQ-Learn}$ breaks in settings with stochastic dynamics,** suggesting that learning Q functions does not transfer as well as learning rewards.
>
> One of the inverse RL baselines that we compared to, **$\texttt{FILTER}$, was shown in [2] to outperform $\texttt{IQ-Learn}$ in MuJoCo experiments with added $p_{tremble}$**. Our Figure 2 experiments were done on the same environments, where $\texttt{FILTER}$ outperformed $\texttt{IQ-Learn}$. Our method, $\texttt{GUITAR}$, matched or outperformed $\texttt{FILTER}$ in those environments.
>
> [1] Garg, Divyansh, et al. "Iq-learn: Inverse soft-q learning for imitation." 2021.
>
> [2] Ren, Juntao, et al. “Hybrid inverse reinforcement learning.” 2024.

---

> > ### Author Response · Authors · 2024-12-03
> > **Follow up**
> >
> > We thank the reviewer for their feedback, and we hope our above responses have addressed any outstanding questions. If there are any other things we can clarify that could help improve your assessment of our work, please let us know.

---

### Official Review · Reviewer_5fqP · 2024-11-04

**Soundness:** 3
**Presentation:** 3
**Contribution:** 2
**Rating:** 8
**Confidence:** 3

**Summary:**

Define misspecification as the expected difference in advantage between an expert policy (taking the optimal action at each state), and a state conditioned policy. Then define an IRL algorithm which searchers for reward using a loss defined by the expected difference in reward under the expert and policy distribution with a divergence constraint to prevent stepping too far, and updating the policy with PSDP. Then, resetting to offline data is analyzed as a complement to resetting to expert data.

**Strengths:**

This work introduces a novel metric for misspecification.

The work offers an interesting perspective on the relationship between IRL and Offline RL

The proposed algorithm shows empirical results that match some of the theoretical claims

**Weaknesses:**

Some of the language is unclear. In particular, the use of sample efficiency and computational efficiency seem to be used interchangeably, but sample efficiency is the more valuable metric. Is computational efficiency even the problem here, and not sample efficiency?

The notation for the advantage function is not formally defined in the work, which makes it slightly more difficult to parse.

While the core definitions for policy completeness error are more or less well defined the GUITAR method itself is not self-contained. It is not obvious what the Bregman divergence or PSDP methods are, and probably more importantly, why these are preferable to other methods. I assume it is because of their theoretical properties, but it is not straightforward to draw the connection between the properties given and those proven in the paper.

The organization of the paper make is more challenging to follow, as the issue of resetting is not the central focus of the work yet intrudes in the middle. If resetting is a core component of the method, then the confusion lies in expressing the connection between resetting and the misspecification problem. Furthermore, this section does not address the original problem of resetting to unreachable states, since the behavior policy may also be unreachable.

The empirical analysis is quite limited. In particular, the performance on more generic tasks is not convincing for this method. In particular the FILTER baseline appears to match or exceed performance in all domains. In several domains, performance is comparable with BC as well. Furthermore, there is no comparison with non-resetting SOTA offline RL methods such as IQL or CQL. Finally, the number of environments for a straightforward, fundamental method is quite limited. In a somain such as this the expectation would be to cover a good portion of at least the deepmind control suite, especially the Franka Kitchen tasks.

**Questions:**

See Weaknesses

---

> ### Author Response · Authors · 2024-11-24
> **Rebuttal**
>
> Thank you for your review. We address the concerns raised below.
>
> ## 1. Ambiguity in language (sample vs computational efficiency)
> > Some of the language is unclear. In particular, the use of sample efficiency and computational efficiency seem to be used interchangeably, but sample efficiency is the more valuable metric. Is computational efficiency even the problem here, and not sample efficiency?
>
> Thank you for highlighting this ambiguity in our language. We clarified our usage of computational and sample efficiency in the revised paper. Sample efficiency is the primary focus of our paper.
>
> ## 2. Advantage function is not defined
> > [omitted due to max character constraints]
>
> We added a definition of the advantage function in Section 2.1.
>
> ## 3. Use of Bregman divergence and PSDP
> > While the core definitions for policy completeness error are more or less well defined the GUITAR method itself is not self-contained. It is not obvious what the Bregman divergence or PSDP methods are, and probably more importantly, why these are preferable to other methods. I assume it is because of their theoretical properties, but it is not straightforward to draw the connection between the properties given and those proven in the paper.
>
> By using a general reduction to an expert-competitive response, our approach can work with any no regret update on the reward function (e.g. follow the regularized leader, online mirror descent, etc) and any expert-competitive response in the policy, which can be computed through any RL algorithm. We added clarification on this point in the paper to make it more understandable to the reader. We specifically select online mirror descent for the reward update and PSDP for the policy update due to their theoretical properties. We clarified this point in the revised paper. The PSDP procedure is outlined in Algorithm 3 in Appendix D, and we added greater explanation for both online mirror descent (including Bregman divergence) and PSDP in Appendix D.
>
> ## 4. Organization of the paper and connection to resets
> > The organization of the paper make is more challenging to follow, as the issue of resetting is not the central focus of the work yet intrudes in the middle. If resetting is a core component of the method, then the confusion lies in expressing the connection between resetting and the misspecification problem. Furthermore, this section does not address the original problem of resetting to unreachable states, since the behavior policy may also be unreachable.
>
> We appreciate your feedback on the paper’s organization. We revised the structure of the paper, adding motivation and context for the issue of resets. In short, our paper considers efficient interactive imitation learning in the misspecified setting. Resets are a key method for improving efficiency in interactive imitation learning. However, the following remain open questions:
>
> 1. Can interactive imitation learning avoid quadratically compounding errors efficiently in the misspecified setting?
> 2. Given that resets are the key method of making interactive imitation learning methods efficient, what is the optimal reset distribution in the misspecified setting?
>
> We then consider settings – both in theory and in practice – where resetting to the expert’s state distribution (as proposed by previous work for the realizable setting) is not the optimal reset distribution.
>
> ## 5. Empirical analysis and comparison to offline RL methods
> > The empirical analysis is quite limited. In particular, the performance on more generic tasks is not convincing for this method. In particular the FILTER baseline appears to match or exceed performance in all domains. In several domains, performance is comparable with BC as well. Furthermore, there is no comparison with non-resetting SOTA offline RL methods such as IQL or CQL. Finally, the number of environments for a straightforward, fundamental method is quite limited. In a domain such as this the expectation would be to cover a good portion of at least the deepmind control suite, especially the Franka Kitchen tasks.
>
> In response to this comment, **we introduce a new experiment with far larger misspecification**. In our new setting, the expert solves the maze by traversing one way through the maze. The learner, however, is unable to take the actions necessary to follow the expert’s course through the maze. (We restrict the learner’s movement by adding additional blocks in the path of the expert.) **Therefore, the learner must solve the maze through an entirely *different route* through the maze.** Under such settings, we see a significant difference between our method (GUITAR) and the IRL baselines (MM and FILTER) and offline baselines (BC).
>
> We did not compare to offline RL methods such as IQL and CQL because, as offline RL methods, they require known rewards. We consider the imitation setting where rewards are unknown, where offline RL algorithms do not work.

---

> > ### Comment · Reviewer_5fqP · 2024-11-25
> > **Response to Authors**
> >
> > I believe that the updates have improved the quality of the paper significantly, and I improved my score accordingly.

---

### Meta-Review · Area_Chair_8369 · 2024-12-20

**Metareview:**

Efficient Imitation under Misspecification

Summary: This paper addresses the limitations of interactive imitation learning (IL) in practical scenarios where perfect imitation of an expert is impossible due to misspecified settings. Unlike prior works focusing on realizable settings, the authors propose a novel structural condition, "reward-agnostic policy completeness," enabling interactive IL algorithms to avoid quadratically compounding errors even when the expert’s policy cannot be perfectly realized. They also tackle the challenge of limited expert data by incorporating suboptimal offline data to enhance sample efficiency. The paper introduces the GUITAR algorithm, which leverages this structural condition and a hybrid reset distribution to improve performance. Empirical results on continuous control tasks validate the theoretical findings, demonstrating that GUITAR outperforms traditional and efficient IL baselines, particularly in scenarios with expert misspecification or finite data constraints.

Comments: We received four expert reviews, with the scores 5, 6, 8, 8, and the average score is 6.75.

The reviewers are generally positive about the algorithmic contributions of this paper and the theoretical analysis. In particular, the reward-agnostic policy completeness error is a novel metric and the corresponding theoretical analysis is novel. The proposed GUITAR algorithm also appears practical for scenarios with limited or imperfect data, making it applicable to challenging domains like robotics and autonomous driving. The paper is well-written, emphasizing both theoretical and practical aspects.

Reviewers have also provided many constructive feedback to strengthen the paper. One major comment is about the lack of strong experimental evidence of setups where the expert's policy is irreproducible/misspecified. The authors have addressed this issue partially during the rebuttal, but more experiments will strengthen the papers. Reviewers have also asked for more discussions on computational complexity and some more clarity in algorithm and experiments discussions. Please update your paper by incorporating all these modifications.

**Additional Comments On Reviewer Discussion:**

Please see the "Comments" in the meta-review.

---

### Decision · Program_Chairs · 2025-01-22

Accept (Poster)